# Degenerate boundaries for multiple-alternative decisions

Sophie-Anne Baker ⬤[1], Thom Griffith ⬤[1] & Nathan F. Lepora ⬤[1] ✉

Integration-to-threshold models of two-choice perceptual decision making have guided our understanding of human and animal behavior and neural processing. Although such models seem to extend naturally to multiple-choice decision making, consensus on a normative framework has yet to emerge, and hence the implications of threshold characteristics for multiple choices have only been partially explored. Here we consider sequential Bayesian inference and a conceptualisation of decision making as a particle diffusing in $n$-dimensions. We show by simulation that, within a parameterised subset of time-independent boundaries, the optimal decision boundaries comprise a degenerate family of nonlinear structures that jointly depend on the state of multiple accumulators and speed-accuracy trade-offs. This degeneracy is contrary to current 2-choice results where there is a single optimal threshold. Such boundaries support both stationary and collapsing thresholds as optimal strategies for decision-making, both of which result from stationary representations of nonlinear boundaries. Our findings point towards a normative theory of multiple-choice decision making, provide a characterisation of optimal decision thresholds under this framework, and inform the debate between stationary and dynamic decision boundaries for optimal decision making.

Choosing between multiple alternatives is a fundamental aspect of animal behavior in natural environments. Such decisions can depend on a reward-dependent task structure that requires a trade-off between speed and accuracy[1–6]. Although significant progress has been made towards understanding the underlying computational principles of binary decision-making, how it transfers to multiple ($n > 2$) choices remains of significant interest[7]. Several strands point towards integration-to-threshold models of multiple-choice behavior comprised of two key mechanisms: integration of accumulated belief informed by sensory evidence represented as the trajectory of a decision variable and a threshold on the belief that triggers a decision[1,5,8–11]. The optimal formulation of these trajectories and boundaries, defined under reward maximization, remains an active topic. Influential models of this type include multiple-choice variants of the drift-diffusion model (DDM)[12], such as the multiple sequential probability ratio tests (MSPRT) and related models[13,14], although the

MSPRT is known only to be asymptotically optimal in the limit of vanishing decision errors[14].

A recent trend in formalizing decision-making is to consider dynamic thresholds that change with time[15–18]. Although these dynamics could conceivably be any continuous process, a popular form is that of a collapsing threshold representing a reducing opportunity cost (over a sequence of trials) or an urgency signal[16]. Collapsing boundaries are known to be optimal for value-based binary-choice tasks[18] and perceptual tasks of mixed difficulty[19], as well as tasks where decisions are subject to a time deadline[20,21]. Evidence for urgency signals comes from observations of the neural activity of the ventral lateral intraparietal (LIP) cortex for four-choice tasks[9]. A recent study[7] applies such dynamic boundaries to multiple-choice decision making with a model that we here refer to as the $n$-dimensional race model ($n$DRM), which uses complex nonlinear boundaries and an urgency signal to maximize reward rate over mixed difficulty trials where the

[1]Department of Engineering Mathematics, Faculty of Engineering, University of Bristol, Bristol, UK. ✉e-mail: n.lepora@bristol.ac.uk

decision-maker's reaction times are freely chosen, but all trials take place within a fixed time period. The $n$DRM model has been shown to replicate behavioral phenomena unique to multiple-choice tasks, such as violations of both the regularity principle and the independence of irrelevant alternatives (IIA) principle.

Almost all of the above-mentioned models of decision-making (except[7]) assume independent, constant-valued decision boundaries for each choice accumulator. For those models, the decision boundaries cannot depend on time or the state of other accumulators, and existing models with interacting accumulators, such as the MSPRT and leaky competing accumulator (LCA) models, limit that interaction to normalization[14] or lateral inhibition across accumulators[22], respectively. However, a core aspect of multiple-choice decision dynamics is that the interaction of accumulators or equivalent decision boundaries may have a non-trivial dependence on the belief over all choices, which can then act as belief-dependent gains on the choice evidence or as nonlinear decision boundaries. In particular, optimal boundaries for general multi-alternative decisions have been shown to be both time-dependent and nonlinear[7]. But how crucial is the precise tuning of the boundary shape to the decision-maker's rewards? Real decision-makers need to transfer limited experience on a decision-making task to decision policies (viewed as boundaries on accumulated evidence) that evolve towards improved outcomes. Therefore, can decision-makers instead find "good enough" boundaries that yield close-to-optimal behavior?

Here we explore this aspect of decision-making, focussing on the consequences of decision boundaries that depend on the (dynamic) state of multiple accumulators. We ask what form the boundaries might take, how they improve decision performance and whether they help explain neural recordings alongside aspects of behavior and generalization. Although here we restrict ourselves to time-independent thresholds, we build upon the findings for time-dependent thresholds to explore the relationship between complexity, nonlinearity, and temporal dynamics. Our particular focus is on the influence of optimality in multiple-choice perceptual decision-making tasks.

Our study adopts a normative approach that focuses first on optimizing complex nonlinear decision boundaries for multiple choices within a parametric subset of possible time-independent thresholds. This setup is followed by examining how these optimized decision boundaries relate to biological implementations and insights. We use a Bayesian framework for decision-making between $n$-choices and then compose a set of complex nonlinear boundary parameterizations that, when optimized, reveal a family of reward-maximizing decision strategies; thus, the boundaries exhibit mathematical degeneracy in belonging to a continuous family of solutions to the optimization problem. Next, we show how these complex nonlinear boundaries lead to implicit dynamics, even though the underlying boundary parameterization is static, meaning that the decision boundaries have an implicit time-dependence despite the boundary parameterization being time-independent. This yields a family of apparent temporal structures that can resemble urgency signals as a direct consequence of having degenerate belief-dependent optimal boundaries for multiple choices. Our results suggest the existence of an unconsidered component in the origin of urgency signals in decision tasks with greater than two choices, such as those recorded in neural populations in area LIP[9]. We demonstrate analytically that our model replicates several multiple-choice phenomena, including the offset in neural activity with the number of alternatives, and violation of both IIA and of the regularity principle, while being compatible with the network model presented in ref. 7 and offering testable predictions.

## Results
### Problem setup and aim
To investigate the form of the optimal decision boundary for multiple choices, we follow the usual convention that choice evidence is

modeled by overlapping normal distributions[23]. Each choice (hypothesis) $H_i$ is represented by a normal distribution with vector mean $\mu_i$ and standard deviation $\sigma_i$. These parameters $\mu_i, \sigma_i$ are defined by the inter-choice discriminability, which is the amount of overlap between choice distributions: the less overlap between distributions $i$ and $j$, the more discriminable the choices and easier the task[24]. We assume equal discriminability between all choices, and so all choice distributions are equivariant with equidistant means, which is achieved by using vector-valued evidence (see Methods). This means that for each decision episode, the "true" hypothesis is equally indiscriminable from all other hypotheses, giving a consistent $n$-alternative forced choice ($n$AFC) paradigm regardless of which hypothesis is chosen.

The integration-to-threshold model samples evidence from the 'true' hypothesis until a decision boundary is reached. Each choice distribution represents possible evidence for that hypothesis, originating from the environment, memory, or noisy sensory processes[18]. At each time step, a sample is taken and inference is performed on the evidence accumulated thus far, generating a decision trajectory. The decision time $T$ is when this trajectory crosses a boundary for a particular choice. If the boundary crossed represents the "true" hypothesis, then zero error $e = 0$ is generated, whereas crossing any other boundary generates a unit error $e = 1$. Usually, integration-to-threshold models rely on scalar evidence with a scalar decision boundary. In our case, the evidence will be a vector with boundaries that are hyper-surfaces in a vector space, which is detailed in the next section.

At the end of a decision episode, when a choice is made, the decision time and error are combined into a single reward. Here, we formulate reward as a linear combination of error and decision time weighted by their associated costs $W_i$ and $c$:

$$r = \begin{cases} -W_i - cT, & \text{incorrect decision} \\ -cT, & \text{correct decision}. \end{cases} \quad (1)$$

This is a standard reward function used in a wide range of past work, for example, ref. 7,25. Unequal error costs $W_i \neq W_j$ induce choice-dependent reward, where hypothesis-dependent error costs are relative to the "true" hypothesis and to each other. For tractability, we will assume all error costs are equal, with the expectation that similar results hold in the unequal cost case but that the analysis will be more complicated. We also consider a constant (time-independent) cost $c$ per time step, assuming stationarity of evidence distributions and evidence accumulation in a free-response task. A challenging aspect of this framework is that reward is highly stochastic due to the random nature of evidence sampling. How then do we define optimality?

In this paper, we come from the view that humans and animals maximize expected reward[19,26–28]. Then the optimum decision boundary maximizes the average reward for a given ratio of costs $c/W$. Monte Carlo simulations of decision trajectories of independent trials, using the formalism outlined above and the evidence inference method derived in the next section, yield reward values for a set of candidate decision boundaries. In general, we find a set of high-dimensional nonlinear, complex boundaries. We will show that these boundaries are consistent with a range of behavioral and phenomenological results along with testable neurophysiological predictions.

### Multi-alternative decision-making as a particle diffusing in $n$-dimensions
In this section, we show that $n$-alternative decision-making can be viewed as a diffusion process in an $(n - 1)$ dimensional subspace of the belief space. This is a perspective that has previously been established (for example, see refs. 7, 14), but we cover this material here to help the reader build intuition and to detail the implications for multi-alternative decisions.

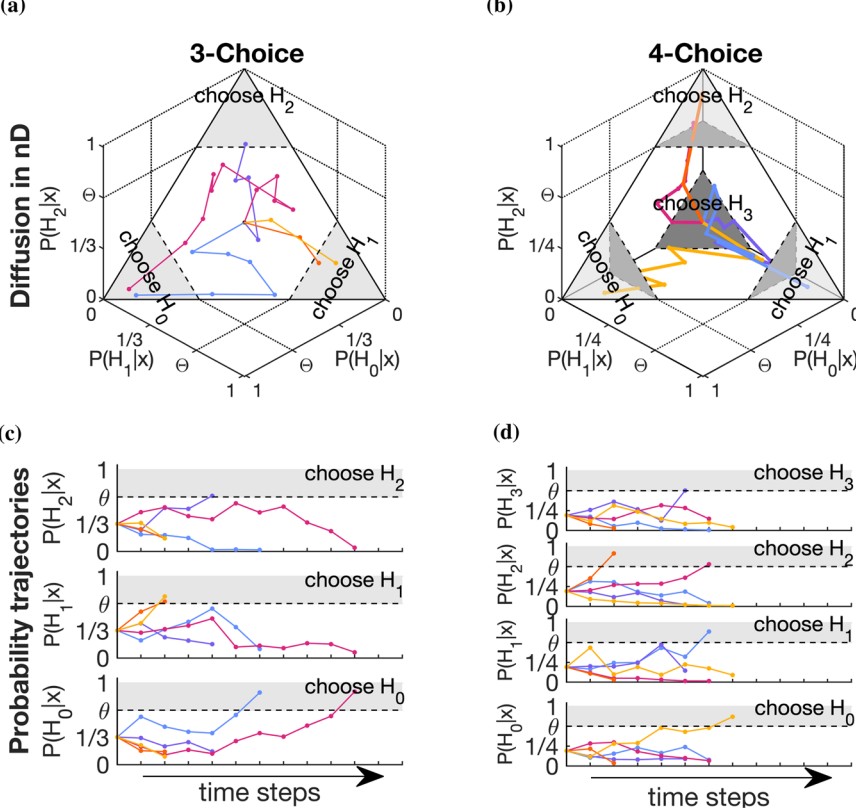

**Fig. 1 | Multiple-choice decision trajectories. c, d** show evidence accumulation, $x$, as independent probability trajectories per choice hypothesis, $H_i$. **a, b** show how these trajectories combine to form $n$-dimensional decision trajectories (colors match between the lower and upper plots). Dashed lines are example "flat" decision boundaries, $\theta$, with shaded gray showing values/areas above the decision boundary in which a decision has been triggered. Notice that the evidence accumulation is constrained to an $(n-1)$D subspace: a plane for three choices (**a**) and a tetrahedron for four choices (**b**). For visualization, (**b**) shows a four-dimensional object embedded into three dimensions projected onto a two-dimensional page; the fourth axis is orthogonal to those shown and "goes into" the page. The import of these different perspectives is that two-choice probability trajectories evolve in time, so decision boundaries can be time-dependent; however, $n$-choice decision trajectories also evolve in space, so decision boundaries can also have spatial dependence and thus a complex high-dimensional structure.

For 2AFC tasks, integration-to-threshold models such as the sequential probability ratio test (SPRT, see Methods), represent the decision trajectory as a particle diffusing in 1D. If we define this dimension on the $y$-axis with the origin corresponding to the point of equal belief for each hypothesis, then positive $y$-values represent greater belief in choice $H_0$ and negative values greater belief in choice $H_1$. If time is represented along the $x$-axis, the decision trajectory takes the form of a random walk over a range of $y$-values, with decision boundaries $y \geq \pm \theta_{0,1}$ for the two decisions 0, 1. This bounded random walk model can be extended to nAFC tasks, the walk taking place in $(n-1)$-dimensions. However, this extension is not straightforward. Firstly, belief in hypotheses $H_{0,1}$ are defined over the positive and negative real numbers of a single dimension, which raises the question of how belief in another hypothesis $H_2$ should be represented. Secondly, the decision boundaries in 1D are well defined as a pair of single bounds ($\theta_0 < y < \theta_1$), but as the belief space extends to $n$-choices, how should the decision boundaries be represented?

To examine these questions, we take a Bayesian sequential inference perspective in which 1D decision variables in models like the SPRT are deconstructed into two decision variables that represent the degree of belief in two hypotheses $H_0$ and $H_1$. By using the sequential Bayesian inference beliefs directly, the positive/negative range for the 1D decision variable is split into two independent axes that represent normalized belief over each hypothesis as a separate decision trajectory, given by the posterior probability $P_i(t) = P(H_i|x(1:t))$ where $x(1:t)$ is the accumulated evidence at time $t$ (see Fig. 1).

The decision variable transformation between the SPRT and sequential Bayesian inference is straightforward (Fig. 1c, d). A decision variable (DV) represents the accrual of all sources of priors and evidence into a quantity that is interpreted by the decision rule to produce a choice[4]. The DV of the SPRT is the log posterior probability ratio[29] and the DV of sequential Bayesian inference is simply the posterior probability. Because sequential Bayesian inference is constrained by $P_0(t) + P_1(t) = 1$, it has the same number of unconstrained degrees of freedom as SPRT. Moreover, the boundary values are equivalent under the DV transformation from SPRT boundaries $\theta_{0,1}$ to boundaries on the posteriors $\Theta_{0,1}$; specifically, the SPRT thresholds are given by the log-odds of the corresponding posterior thresholds in sequential Bayesian inference[4],

$$\theta = \log(\Theta/1 - \Theta). \qquad (2)$$

Now, the key point is that sequential Bayesian inference applies to an arbitrary number of choices and so holds for general $n$AFC decision-making[14]. Figure 1c, d illustrate this for 3- and 4-choice tasks, respectively, with the dashed lines representing flat decision boundaries $P_i(t) > \Theta_i$. Individual probability trajectories for each choice correspond to the coordinates of the overall decision trajectories (Fig. 1a, b), interpreted as a particle diffusing in $n$D. Sequential Bayesian inference forms an orthogonal coordinate system for each probability trajectory (Fig. 1c, d) as components of the $n$-dimensional decision trajectory (Fig. 1a, b).

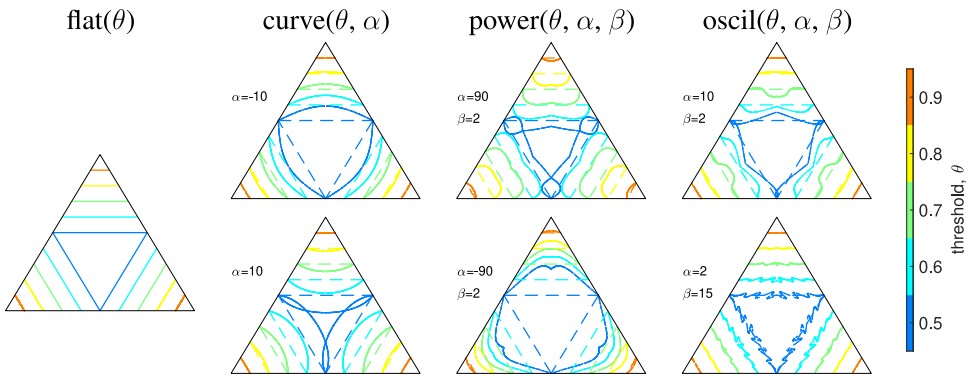

**Fig. 2 | Example boundaries for three choices.** Each column belongs to the indicated parameterization and each row (apart from the flat case with just one parameter) shows two opposite extreme examples. Colors indicate the edge-intersection parameter $\theta$, and parameter values $\alpha$ and $\beta$ are indicated as appropriate. Notice the diverse range of curves to explore spatially-dependent optimal boundaries. Comparison with the flat($\theta$) parameterization is shown by the dashed colored lines.

There are geometric implications of using sequential Bayesian inference as coordinates for $n$-dimensional decision trajectories (Fig. 1a, b). Although the decision trajectories have $n$ probability-coordinates, $\mathbf{P}_n$, they are constrained such that $\sum_i P_i(t) = 1$; therefore, the decision trajectories populate $(n-1)$D simplices. For example, 2AFC decision dynamics are represented as a particle constrained to have $P_0(t) + P_1(t) = 1$, which is the 1D line $P_1 = 1 - P_0$ on a 2D ($P_0$, $P_1$) plot of the beliefs. It follows that 3AFC dynamics take place on a 2D plane (Fig. 1a) and 4AFC dynamics in a 3D tetrahedron (Fig. 1b) and so forth. Note that if any hypothesis has zero probability $P_i(t) = 0$, then the space in which the decision trajectory evolves collapses to the remaining non-zero directions. For example, each face of the 4-choice tetrahedron in Fig. 1b is a combination of three choices with non-zero probabilities and each edge a combination of two such choices.

As a result, decision boundaries are $(n-2)$-dimensional objects in $n$-dimensional probability space. So for $n > 2$ choices, boundaries can have spatial dependence with respect to the decision space visualized in Fig. 1a, b. For 2AFC tasks, decision boundaries are points on a 2D line, which are simply the transformed boundaries (equation 2) of the standard two-choice integration-to-threshold model. Likewise, for 3AFC, the decision boundaries are lines on a plane (Fig. 1a, dashed lines) and for 4AFC are planes within a tetrahedron (Fig. 1a, dark gray planes). The example boundaries shown are flat with a constant decision threshold in each dimension. An interesting consequence is that high-dimensional boundaries can have a nonlinear structure as a function of the $n$-dimensional beliefs $\mathbf{P}$. Then, the linear 3AFC boundaries (Fig. 1a, dashed lines) generalize to curves and the planar 4AFC boundaries (Fig. 1b, dark gray planes) generalize to curved surfaces.

Curved decision boundaries have been shown to perform optimally on 3AFC tasks for free-response, mixed-difficulty trials[7]; however, it is not known how important the precise shape of that boundary is for maximizing reward. Here we ask whether there are other complex boundary shapes that improve performance over the flat boundary case and whether the greater freedom to choose nonlinear boundaries has other consequences for decision-making.

**Multi-dimensional decision boundaries can be complex**
To investigate the importance of boundary shape for reward maximization, we define a subset of possible boundaries using some specific spatial parameterizations that provide diverse sets of nonlinear boundaries. These parameterizations are constrained such that: (I) each boundary $\theta_i$ intersects with each edge leading away from the point $P_i = 1$, and likewise intersects with each (hyper)plane leading away from the said point (e.g., each colored boundary intersects with two edges in Fig. 2); and (II) assuming symmetric error costs $W_i = W_j$ for simplicity in equation (1), the boundaries remain symmetric under permutations $P_i \leftrightarrow P_j$ (e.g., all boundaries have the reflectional symmetries of the outer equilateral triangle in Fig. 2).

These constraints can be used to derive a general boundary parameterization comprising a shape function and tuning parameters (Fig. 2). A general boundary parameterization $F(\mathbf{P}(t); \theta, \alpha, \ldots)$ takes the probability vector $\mathbf{P}(t)$ as an input, along with an edge-intersection parameter $\theta$ and shape parameters $(\alpha, \ldots)$ to give a decision rule:

$$P_i(t) > F(\mathbf{P}(t); \theta, \alpha, \ldots). \tag{3}$$

The resulting complex decision boundary has an amplitude parameter $\alpha$ and some additional shape parameters. To make our investigation tractable, we limit our parameterization to one additional parameter, $\beta$ (e.g., a frequency in the oscillating case). For simplicity, we select four distinct forms of $F$ that we call flat($\theta$), curve($\theta, \alpha$), power($\theta, \alpha, \beta$), and oscil($\theta, \alpha, \beta$), examples of which are shown in Fig. 2 and all of which contain the flat boundary as a particular instance (see Methods for the full forms and a mathematical derivation). Within these parametric subsets, the optimal decision boundaries are determined by optimal values of $\theta$, $\alpha$, and $\beta$.

Some example boundary parameterizations illustrate the range of possible boundary features and how they extend to multiple-choice decision tasks (Figs. 2 and 3). Each parameterization is a scaling of a flat boundary (Fig. 2, left column) denoted as the flat($\theta$) function, such that: (I) The function curve($\theta, \alpha$) is the simplest parameterization of interest, with $\alpha$ the amplitude of the curve (Fig. 2, second column). (II) The function power($\theta, \alpha, \beta$) has an additional parameter $\beta$ that modulates a double curve or forms a central peak (Fig. 2, third column). (III) The oscillatory function oscil($\theta, \alpha, \beta$) is a cosine with amplitude $\alpha$ and frequency $\beta$ (Fig. 2, fourth column). We have chosen these parameterizations so that if $\beta = 0$, we recover the curve parameterization, and if $\alpha = 0$, we recover the flat parameterization. These parameterizations apply to any number of choices $n \geq 2$, with examples of the curve and oscil functions for 4AFCs shown in Fig. 3a, b respectively. Note how these decision boundaries intersect with each 3AFC plane: each face in Fig. 3a matches a panel in Fig. 2.

Overall, we have constructed a set of permutation-invariant, nonlinear decision boundaries that we will use as candidate functions to explore optimal decision rules. This raises the question of which parameter values give optimal boundaries within these parametrized subsets of boundaries.

**Complex decision boundaries are consistent with the speed-accuracy curve**
It is well established that humans and animals generate speed-accuracy trade-off (SAT) curves during decision-making experiments[6,9],

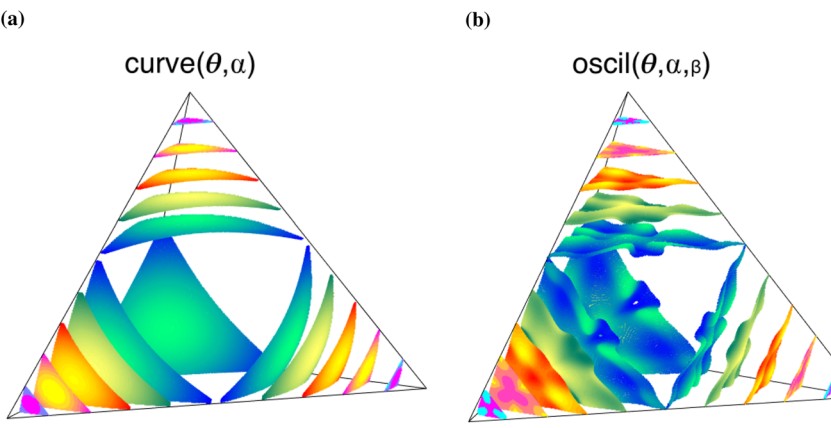

**Fig. 3 | Example boundaries for four choices.** Example decision boundaries are given by the curve parameterization (**a**) and the oscil parameterization (**b**). The colors/levels are different edge-intersection parameter values $\theta$, using the same values as those in Fig. 2, where (**a**) is the $N = 4$ equivalent of the first curve example. Shading is used to visualize the shape of the boundary.

showing the mean error against mean decision time, where each point on the curve can be accorded a cost ratio $c/W$ of time to errors (equation (1)). For 2AFC tasks, the trend is such that as the value of $c/W$ increases, speed is favored over the accuracy, and so the mean decision time decreases with a compensatory increase in mean decision error. This trade-off is instantiated by the decision rule (learned for each $c/W$ value) with the SAT curve implicitly parameterized by decision boundary parameters. For 2AFCs, this is a single parameter: the flat boundary threshold $\theta$[30,31]. For $n$AFCs with $n > 2$, these are complex boundary functions (equation (3)) with sets of parameters ($\alpha$, $\beta$, ...). If the SAT curve is truly a curve, rather than a region, then multiple parameter combinations would give the same SAT, since a curve requires just one implicit parameter.

To examine the SAT curves for each parameterization, we optimized the boundary parameters for a range of cost ratios $c/W$ and then plotted the resulting accuracies and decision times (Fig. 4). Optimal parameters $\theta$, $\alpha$, $\beta$ were found by stochastic optimization over the reward landscapes, using Monte Carlo simulation over a grid search of parameters to generate a visualization of the reward landscapes (Fig. 5a–c). As rewards are highly stochastic, smoothed landscapes were averaged over 100,000 decision trajectories. Optimal parameters were extracted by taking all points with a mean reward for which $r > r_{max} - \delta\sigma$, where $r_{max}$ is the maximum mean reward of the noisy landscape, $\sigma$ is the spread of rewards, and $\delta = 0.02$ is a small parameter (see Methods for details). Since the variation in expected reward is so small as to be negligible for a real decision-maker with limited capacity for sampling rewards from the environment, we define these boundaries as constituting a set of "good enough" boundaries that are in practice as effective as a true optimum within each parameterization.

The boundary functions curve, power, and oscil produce smooth, well-defined SAT curves (Fig. 4) that resemble the relationship between speed and accuracy for optimal rewards found experimentally[2,31]. Despite the wide range of boundary shapes they describe, all three parameterizations produce nearly identical SAT curves, as confirmed by overlaying the average over all three cases (Fig. 4, black dashed curve, all panels). Flat boundaries ($\alpha = 0$) are contained within these parameterizations, and so the SAT curves for all three boundary functions closely resemble the flat-boundary case. One difference between the three cases is their relative spreads: the curve($\theta, \alpha$) parameterization yields the tightest SAT curve, followed by the oscil($\theta, \alpha, \beta$) SAT curve, and finally, the power($\theta, \alpha, \beta$) SAT curve is the thickest. One might therefore consider that the curve parameterization qualifies as the 'best' SAT curve, which is consistent with previous work since it describes the shape of the optimal boundary found for the 3AFC case in refs. 7,14. However, all parameterizations

closely follow a single mean SAT curve (black dashed lines), so a wide range of boundary characteristics give near-optimal decisions. Moreover, each value of $c/W$ has multiple points spread along the same curve, so the SAT can be satisfied by multiple optimized boundaries even within the same parameterization.

## A degenerate set of decision boundaries yield close-to-optimal expected reward

So far, we have seen that optimized nonlinear decision boundaries generate well-defined SAT curves that remain similar in three parameterizations curve, oscil, and power. Next, we analyze the optimized boundaries by direct inspection of the reward landscapes and their position on the SAT curve.

Inspection of the 3AFC reward landscapes for the curve parameterization reveals that the region with mean rewards within $\delta$ of the maximum $r_{max}$ extends across the parameter space (Fig. 5, black lines). As $c/W$ increases and the maximum reward $r_{max}$ decreases (panels a–c), this acceptance region sweeps towards $\theta = 1/3$ and the flat boundary $\alpha > 0$ and becomes more dependent on $\alpha$ in addition to $\theta$. These acceptance regions correspond to sets of optimized parameter combinations and so specify families of decision boundaries that all maximize reward within a small variance.

For closer scrutiny of the optimal region, five sections of the reward landscapes for 3AFC curve($\theta, \alpha$) parameterization are shown in Fig. 6. These sections correspond to values $\alpha = \{-20, 10, 0, 10, 20\}$ (including the flat boundary $\alpha = 0$) to provide a detailed look at the reward landscape peaks with 100-fold more samples of $\theta$ than in Fig. 5. Evidently, the peak changes with cost ratio $c/W$ and $\theta$ (Fig. 6, right column). This analysis leads to the observations: (I) As $c/W$ increases, the reward landscape maximum covers a broader range of $\theta$ values (section peaks separate) and appears to acquire slope (peaks diverge in height, with a higher-to-lower pattern). (II) The spread of the average rewards at the peak of each section overlaps (red dashed lines), decreasing as $c/W$ increases but not diminishing to zero. (III) Extracting near-to-optimal parameters by taking all points within a small $\delta = 0.02$ range of the peak average reward yields a set of near-optimal decision boundaries over a broad range of $\theta$ and $\alpha$ values (black points in Fig. 6).

These three observations all support the effective degeneracy of optimized decision boundaries within the parameterizations. Observation (I) shows that for small $c/W$, the underlying structure of the reward landscape appears to degenerate with sections almost entirely overlapping (Fig. 6, top right). As $c/W$ increases, a shallow structural maximum becomes apparent (Fig. 6, bottom right). Observation (II) shows significant overlap in the close-to-optimal region even across the apparent structural maximum (Fig. 6, bottom right). Observation

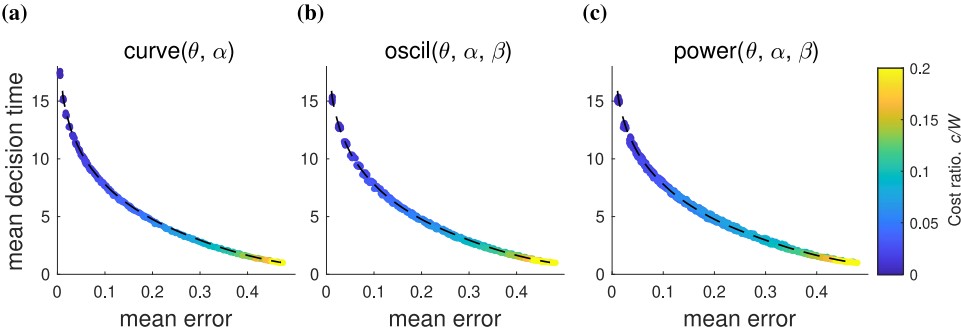

**Fig. 4 | Speed-accuracy trade-off curves for 3AFCs optimized over all parameters.** Colors indicate the cost ratio $c/W$ (blue through to yellow) for **a** the curve parameterization, **b** the power parameterization, and **c** the oscil parameterization. The average SAT curve (dashed line) is a piecewise-linear curve from the mean decision time and error over all three parameterizations for each $c/W$ value (see methods). All parameterizations (**a**–**c**) give regions of speed-accuracy trade-off that are tightly gathered around the mean SAT curve and depend smoothly on the cost ratio $c/W$.

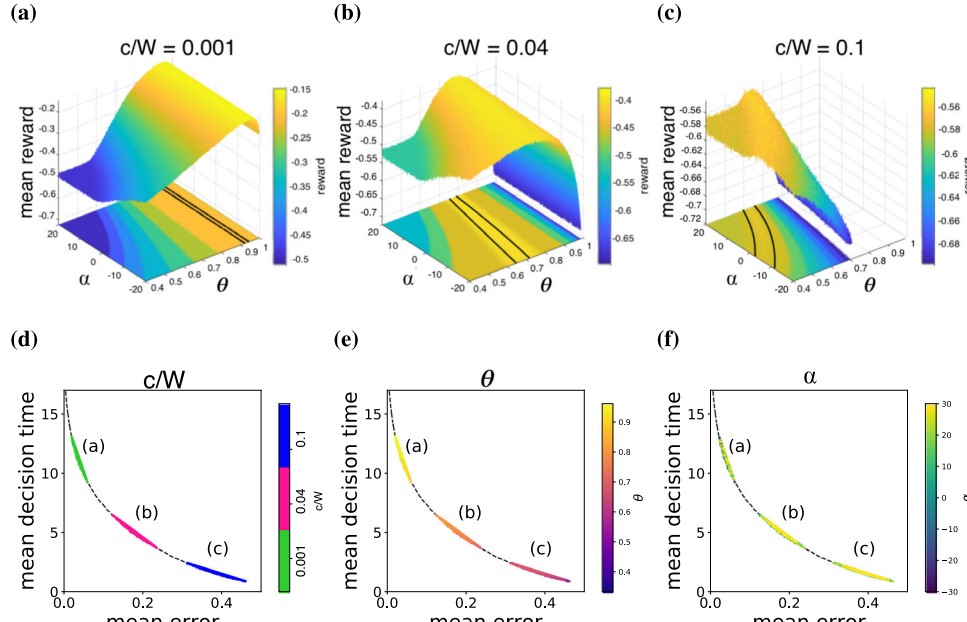

**Fig. 5 | Reward landscapes and effective degeneracy of optima. a**–**c** show example reward landscapes over parameters of the curve($\theta$, $\alpha$) decision boundaries for cost ratios $c/W$ of 0.001, 0.04, and 0.1 for 3AFCs. The regions of maximal reward are outlined (black lines for $\delta = 0.02$; see text) within a projection of the mean reward onto a horizontal plane, which reveals an extended region of nearly-optimal decision boundaries. **d**–**f** show the speed-accuracy trade-off for decision boundaries encompassed in the maximal region of **a**–**c**, with the black dashed line showing the mean SAT curve from Fig. 4. Notice the "spread" of SAT values for the same cost ratio and similar values of $\theta$. There are apparently two types of degeneracy in the set of optimized decision boundaries: boundary shape (**a**–**c**) and SAT (**d**–**f**).

(III) shows directly that there is an effective degeneracy. One could question whether different sections through the reward landscape would change these observations, as Fig. 6 depends on the range and discretization of $\alpha$. Our range of $\alpha$ covers the entire range of boundary shapes shown in Fig. 2, including flat boundaries, and because of the gradual variation across sections, we would not expect further structure from a finer discretization. We also expect that using more Monte Carlo samples for each cross-section would not change the results, as the means and spreads shown in the sections in Fig. 6 appear to be good estimates of the distributions of average rewards (e.g., by their smooth variation with $\theta$ and unitary maxima).

The close-to-optimal set of boundaries produces a range of speed-accuracy trade-offs (Fig. 5d–f). In 2AFC decision-making, each point on the SAT curve is a unique optimal boundary specifying a unique trade-off for a given value of $c/W$. This raises the question: for nAFCs with $n > 2$, what are the range of points on the SAT curve given by the set of close-to-optimal boundaries? Fig. 5d shows the mean decision errors

against mean decision times for all close-to-optimal boundaries found in the 3AFC landscapes from Fig. 5a–c. In each case, an effectively-optimal reward is achieved by a broad range of SATs (Fig. 5d) rather than a tight group around a single SAT.

How can a small range of reward values produce a broad range of speed-accuracy trade-offs? The breadth of speed-accuracy trade-off values produced by complex decision boundaries is explained by a range of near-optimal threshold parameter values. Then, given cost values $W$ and $c$, the rewards

$$\mathbb{E}(r_{max}) = -W\mathbb{E}(e) - c\mathbb{E}[T], \quad e = \{0,1\}, \quad (\text{correct/incorrect decision})$$
(4)

have two degrees of freedom for each value of $\mathbb{E}(r_{max})$ in trading off the expected error $\mathbb{E}(e)$ and expected decision time $\mathbb{E}[T]$. Thus, the same expected reward may be attained by boundaries with different combinations of $\mathbb{E}(e)$ and $\mathbb{E}[T]$. Hence, the set of close-to-optimal

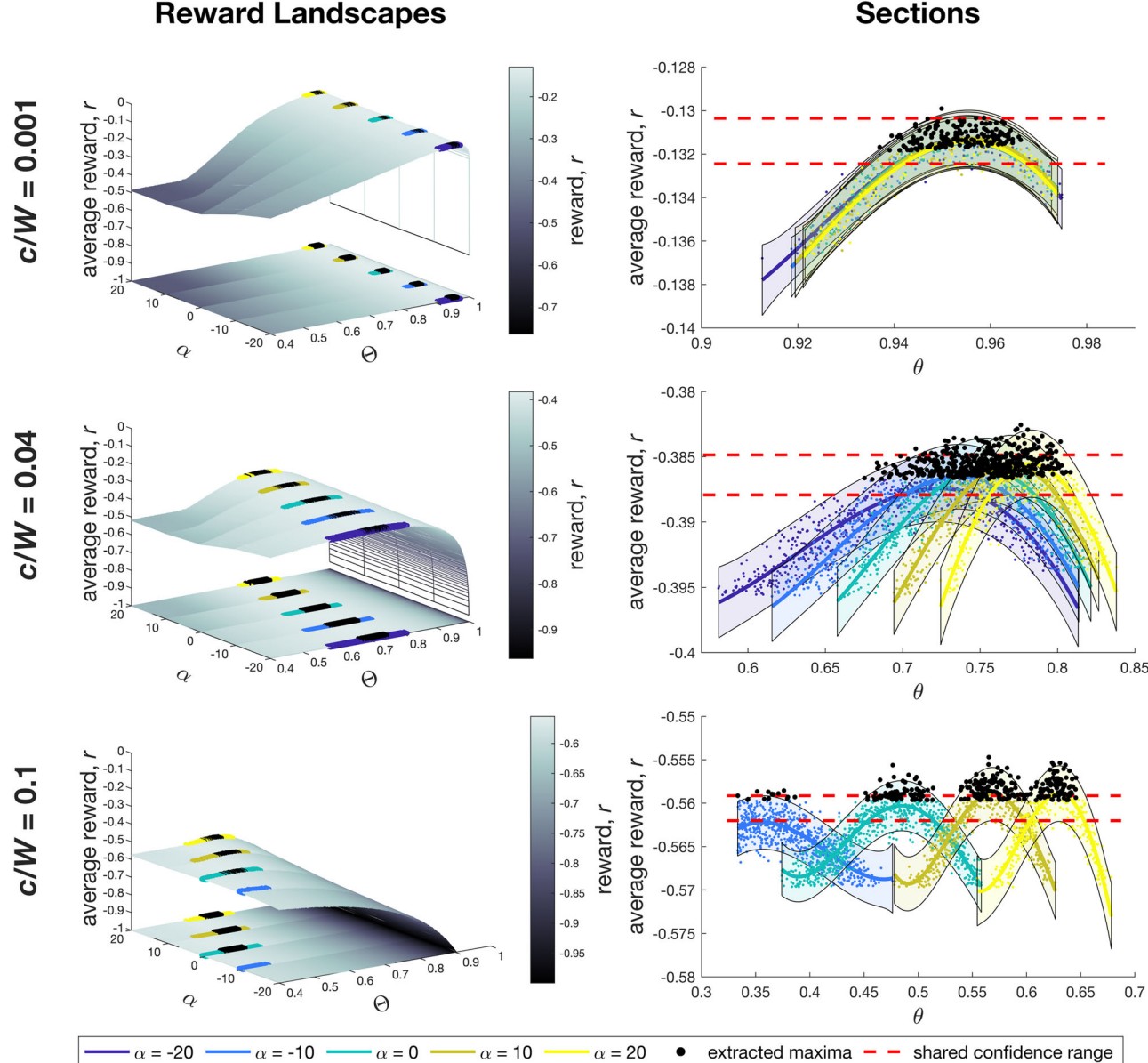

**Fig. 6 | Sections through reward landscapes.** Five sections through the reward landscapes from Fig. 5a–c for the 3AFC curve($\theta$, $\alpha$) parameterization are shown in the left-hand column of panels for $\alpha = \{-20, -10, 0, 10, 20\}$ (colors in legend). Note that the $\alpha = 0$ (cyan) sections are flat decision boundaries. Average rewards across the sections are shown in the right-hand column with their mean (solid lines) and spread (transparent area), both are estimated with Gaussian Process regression; 95% confidence bounds. The red dashed lines show the overlap between the spread of all sections at their peak mean rewards. Maximal regions lie within a small fraction ($\delta = 0.02$) of the standard deviation of the peak reward, giving a degenerate set of decision boundaries with close-to-optimal rewards (black points).

**Effectively-optimal decisions with different boundary shapes**
What are the characteristics of the nonlinear decision boundaries in the near-optimal set? If the decision boundaries were qualitatively similar in shape, then learning the particular boundary shape would be important for maximizing reward. Conversely, if the boundary shapes are qualitatively different, then learning the precise boundary shape would not be critical, and the emphasis on optimality would shift to the inference and normalization processes for multiple-choice decision-making.

Figure 5e shows that for each cost ratio $c/W$, the edge-intersection parameter $\theta$ seems to be the main determinant of optimality, as is visible in the homogenous values of $\theta$ within each region (a–c). In

contrast, the shape parameter $\alpha$ takes heterogenous values within each region (a–c) in Fig. 5f. For every cost ratio $c/W$, the entire explored range of $\alpha$ is represented in the optimal set, whereas there is a narrow range of $\theta$ that varies with $c/W$. The flat-boundary case ($\alpha = 0$) is optimal for each of the degenerate sets, with the optimal $\theta$ then a single value that lies within the broadened range when $\alpha$ is non-zero. Thus, for all $c/W$, there are many SATs near each parameter value, and in turn, each SAT instance is close to many different parameter values (Fig. 5e, f).

Therefore, it appears that close-to-optimal multi-alternative boundaries are possible with significant modulation of the flat-boundary case. The close-to-optimal set contains a broad range of parameters, giving a diverse set of boundary shapes (examples in Fig. 7). This supports the notion that learning the precise boundary shape is less important for making effective optimal decisions.

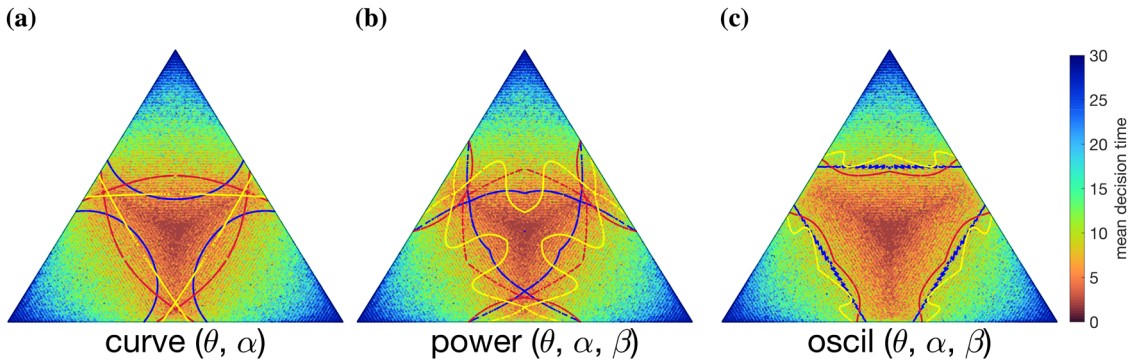

**Fig. 7 | Examples of qualitatively-different degenerate decision boundaries overlaid on the mean decision times for flat boundaries.** Here we show the curve (**a**), power (**b**), and oscil (**c**) parameterizations for 3AFCs. All plotted boundaries maximize reward to the same value (for the same cost ratio $c/W = 0.04$) despite having no other common characteristics. **a**–**c** show a map of the mean decision time over the belief space, with low decision times at the center increasing towards the edges. Decisions, and hence decision times, only manifest at decision boundaries, so to visualize the variation over the whole belief simplex, we generated an approximation of the mean decision time distribution using incrementally-increasing flat decision boundaries.

## Mean error and decision time vary along the optimal decision boundaries

Every point on an extended boundary for $n$AFC decision-making with $n > 2$ has an associated error and decision time distribution (see Fig. 7 for a color map of the mean decision time for flat boundaries). In contrast, the 2AFC decision thresholds are points on a line in the space of 2D belief vectors **P** from (0, 1) to (1, 0). Each choice on an extended boundary is a single point with an error and decision time distribution. Spatially-dependent error and decision time distributions that vary along the decision boundary are a consequence of having a multi-dimensional belief vector that can vary over a decision boundary embedded as a curve or surface in the higher-dimensional belief space.

Hence, the close-to-optimal set of boundaries have different ranges of mean decision times and mean errors but practically indistinguishable expected rewards. If the under-determinism in the reward structure were eliminated (e.g., by also minimizing mean error or mean decision time), then the set of reward-maximizing decision boundaries would contract and SATs narrow.

Interestingly, decision times vary even along flat boundaries (Fig. 7a, yellow lines; decision times shown by background color), and so even the simplest case of crossing a single threshold is complicated for multiple alternatives.

## Implicit dynamics of optimal decision boundaries support both static and collapsing thresholds

The point where a decision trajectory crosses a high-dimensional decision boundary is a belief vector that has a corresponding mean error and mean decision time. Because all of these quantities can vary along a nonlinear boundary, there can appear to be non-trivial dynamics in the decision 'threshold' if it is instead interpreted as a unitary value rather than as a boundary function. Here we refer to this property as implicit dynamics because it originates in the boundary shape rather than from an explicit time-dependence of the threshold. In Fig. 7, the boundary overlays a gradient coloring representing decision time, making explicit the non-trivial relation between decision time and belief of a decision. In this sense, we uncover temporal 'dynamics' implicit in the static, complex, and nonlinear decision boundaries for multiple choices.

There has been much debate over whether temporally-dynamic decision thresholds give a better account of 2AFC experimental data than the static thresholds of SPRT[15,16,18]. The assumption of fixed (constant-valued) thresholds has been called into question, with collapsing thresholds gaining popularity, which are sometimes interpreted as urgency signals[9,19,30,32]. From an optimality perspective, collapsing thresholds are more appropriate for repeated free-response trials of mixed difficulty and for those with deadlines[19,21], whereas static thresholds are appropriate for single free-response trials without deadlines and repeated free-response trials of known difficulty; however, static thresholds are not adequate for single free-response trials of mixed, a priori unknown difficulty[25]. Models with collapsing thresholds have been shown to reduce the skew of error and decision time distributions in some experimental tasks[16], and urgency signals proposed to account for increased firing rates in the LIP brain region of macaques during trials in which accumulated evidence (encoded as neuronal firing rates) is unchanging[10,11,30]. How multiple-choice decision boundaries relate to this debate is thus of interest.

To investigate the relationship between implicit decision threshold dynamics discussed above and spatial nonlinear decision boundaries, we transform the decision variables to a form that gives a temporal structure in the threshold as a consequence of the extended static boundaries. The boundary beliefs are sorted by decision time, averaging over boundary values with identical decision times. These then appear as time-dependent decision boundaries applied to evidence (Fig. 8), which for display purposes, we represent using the log-odds (equation 2).

These dynamic decision thresholds have a range of temporal structures for each cost ratio $c/W$, separating naturally into three categories: increasing (Fig. 8, top row), collapsing (middle row), and static thresholds (bottom row). These categories appear to correlate with the shape of the decision boundary: increasing thresholds with convex boundaries (e.g., Fig. 7a, dark blue curve), decreasing thresholds with concave boundaries (e.g., Fig. 7a, red curve), and static thresholds with flat boundaries (e.g., Fig. 7a, yellow line). This correlation seems to originate in an increase in decision time as the belief moves away from equality between choices (Fig. 7a–c, shading). All things being equal, decisions that terminate with higher beliefs tend to be more accurate, whereas decisions terminating with a low belief of the choice tend to be less accurate. Therefore, experimental observation of implicitly dynamic decision boundaries (including increasing thresholds) could be due to time-dependent accuracy plots of data from individual subjects[33].

We emphasize that while these implicit threshold dynamics look like temporal dependence, they are, in fact due to the spatial structure of the decision boundary. Flat decision boundaries can therefore be interpreted as a special case where the boundary on the belief in one choice to cross its threshold is independent of the beliefs of the other choices; however, even then, the mean decision time and mean error have a spatial structure along these boundaries (Fig. 7).

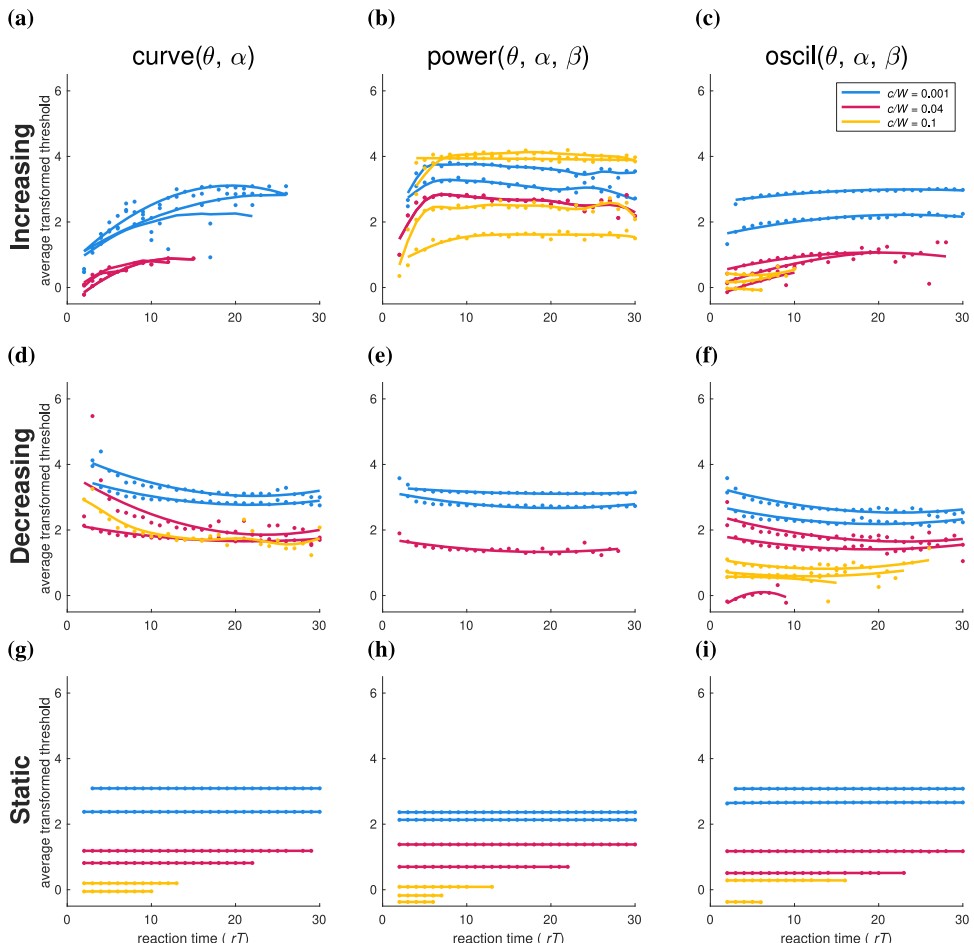

**Fig. 8 | Implicit threshold dynamics.** Columns separate results by curve, power, and oscil parameterization for 3AFC decision-making. The rows show examples of implicit dynamics categorized as increasing (**a**–**c**), decreasing (collapsing, **d**–**f**), and flat (static, **g**–**i**). Colors indicate the cost ratio $c/W$ for which the thresholds are optimized. Note that the transformed threshold refers to taking a log-odd scale for better visualization (equation 2). Therefore, nonlinear static, high-dimensional decision thresholds exhibit implicit dynamics with a correlation between those dynamics and the boundary shape.

## Comparison to current models, interpretations, and predictions

There are few normative approaches to modeling multiple-choice decision-making, with the recent study of ref. 7 the state-of-the-art in using an evidence accumulation vector accumulated in a race model with a boundary in $n$-dimensional space. Using dynamic programming, they find optimal decision boundaries for free-response, mixed-difficulty trials are nonlinear and collapse over time. Clearly, their $n$DRM is closely linked to the model presented here, but there are key differences: the models have a different evidence structure, trial structure, and optimization process, which leads to diverging perspectives on the nature of multiple-choice decision boundaries. In the following, we describe how these perspectives can be reconciled, and in doing so gain a broader understanding of normalization and inference in multiple-choice decision-making.

In comparison with the nDRM, one difference between our model and that in ref. 7 is in how they use Bayesian inference to accumulate evidence. Although the $n$DRM is derived from Bayes' rule, evidence is accumulated linearly and inference values (the posterior) are used indirectly to calculate the expected reward[7]. Conversely, our model, which focuses more on the details of nonlinear decision boundaries, uses the posterior from Bayes' rule as the accumulated evidence. In this respect, our model has less biophysical realism that the $n$DRM because there is little supporting evidence for the brain representing posterior beliefs directly as probabilities. Instead, studies point towards the brain employing indirect representations from which beliefs can be inferred[33,34]. However, we describe below how either

view of evidence representation is compatible with two of the main results of the present paper.

Firstly, evidence accumulation takes place on an $n - 1$-dimensional simplex in our model, as in Fig. 1a, b ($n = 2$ a curve; $n = 3$, a plane). Similarly, reward maximization in the $n$DRM results in evidence accumulation perpendicular to a diagonal equidistant from all evidence-component axes, and so the space collapses to $n - 1$ dimensions. This subspace appears to be a scaling of the posterior simplices shown in Fig. 1a, b, which is simply a normalization of the evidence accumulation (see Methods). In consequence, both models agree that normalization of the evidence is a key component of multiple-choice decision-making, which is imposed by using probability values directly here and emergent in the $n$DRM. Further, the decision boundaries in both models exist in that same subspace, and so agree as to the expected dimensionality of neural population activity during evidence accumulation (see later section on Predictions).

Secondly, our model results in a set of close-to-optimal decision boundaries that contains drastically different boundary shapes, which we now argue is compatible with the $n$DRM. Tajima et al. use a network approximation to the optimal decision rule to separate the boundary components into a race model with nonlinear and dynamic (collapsing) boundaries, both with normalization, and evaluate their performance under reward maximization[7]. Their model performance is best in the presence of internal variability (adding noise), where omitting boundary dynamics performs similarly or slightly better than the combination of nonlinearity, dynamics, and normalization together.

From our perspective, these distinctly different types of boundaries could be interpreted as members of a set of close-to-optimal boundaries since both types result in effectively maximal rewards.

The effect of evidence normalization is also important. Both the model presented here and the $n$DRM rely on evidence normalization as key to model performance. In the model here, the normalization of evidence follows from using posteriors, whereas in the $n$DRM it follows from a projection of the accumulated evidence onto the $n-1$-dimensional subspace described above. This suggests a major influence on optimality for multiple-choice tasks, yet an assessment of the contribution of normalization alone on performance is absent in ref. [7]: in the $n$DRM, normalization is not separated from the nonlinearity of the decision boundaries. For this reason, and because[7] uses mixed difficulties which demand collapsing boundaries, comparing the performance to flat boundaries for the $n$DRM is not comparable to the flat boundaries examined here. Normalization appears integral to optimal multiple-choice decision-making, and the magnitude of its influence may offer an explanation for degenerate optima: it appears that boundary shape has a lesser influence on optimality when the evidence is normalized, which would benefit learning and generalization as a general neural mechanism for context-dependent decision-making[35].

Evidence normalization is also a mechanism that satisfies a number of physiological constraints. Represented by the range of neural activity, normalization satisfies: (I) Biophysical constraints of the range of activity of neural populations – the firing rate of biological neurons cannot be negative and cannot exceed a certain level due to their refractory period[13]; (II) neural recordings of decision-making tasks show that a decision is triggered when neural activity reaches a stereotyped level of activity[9], and (III) as the number of options increases, so too does the processing and representation of multivariate evidence accumulation and the relative belief over these options. This, together with the wide-ranging influence of normalization on optimality discussed previously, leads us to consider normalization as an integral part of the evidence accumulation process.

We now show that normalization, here in the form of a posterior representation of the evidence, can explain some 'irrational' behaviors: (a) the decrease in offset activities in multi-alternative tasks; (b) violation of the independence of irrelevant alternatives, and (c) violation of the regularity principle[9,36–39]. These behaviors are outcomes of three properties of normalization when adding (for example) a third option $P_2$, where the currently-held beliefs are $P_0$ and $P_1$ such that $P_0 + P_1 = 1$; then the new normalized probabilities are

$$\tilde{P}_i = \frac{1}{1 + P_2} P_i. \qquad (5)$$

This has the effect of increasing the minimum distance $d_T = T - P_i/(1 + P_2)$ from each choice belief to a boundary $T$, decreasing the distance $d = P_i/(1 + P_2) - 1/3$ from the flat prior, and reducing the difference in belief values supporting each alternative

$$\Delta_{i,j} = \frac{1}{1 + P_2} |P_i - P_j|, \qquad i \neq j. \qquad (6)$$

All of these quantities decrease as the belief value of option $P_3$ increases, which has the following consequences.

First, we consider (a) a decrease in offset activities in multi-alternative tasks. Multiple studies show that the initial average neural activity ("offset") encoding evidence accumulation decreases as the number of options increases[9,36]. Support for this offset behavior is given in ref. [7] for the network model of the $n$DRM by introducing it directly into the reward maximization as the number of options is increased. However, assuming that evidence accumulation uses the posterior directly, as in our model, then for each unit increase in the number of choices, the average belief per choice decreases by $1/n(n+1)$; i.e., the decrease in offset activity is a direct consequence of evidence normalization.

Next, we consider (b) the violation of IIA. The independence of irrelevant alternatives (IIA) recurs in many traditional rational theories of choice[40,41]. It asserts that the presence of an 'irrelevant' option should not affect the choice between "relevant" options (e.g., adding a low-value choice to existing higher-value choices)[42–44]. Violation of this principle has been shown across behavioral studies in both animals and humans[37]. This behavior is replicated in[7] using a network approximation of the $n$DRM with added noise during evidence accumulation and is attributed to their use of divisive normalization. In our model, the violation of IIA is explained by the representation of evidence accumulation as the belief vector (equations (5, 6)): as the belief of the third option ($P_2$) increases, the belief values of other options are reduced, which necessitates further evidence accumulation before making a decision than would otherwise have been required. Further, the difference in belief values supporting both high-valued options is decreased (equation (6)), so more evidence is needed to choose between these options also. Requiring additional evidence accumulation, and so increased difficulty in choosing between the two high-valued options, is exhibited by longer decision times and/or higher error rates, as in behavioral studies.

Last, we consider (c) the violation of the regularity principle. The inverse of IIA violation, the regularity principle, says that adding extra options cannot increase the probability of selecting an existing option, and has been found to be violated in behavioral studies[38,39]. This is simulated in[7] using the same network approximation of the $n$DRM as for the violation of IIA. We also find that violation of regularity is congruent with IIA violation: adding a third option reduces the belief value of the original options (equation (5)) while also reducing the difference in belief values (equation (6)).

Our next consideration is related to network models of decision-making. In the $n$DRM, evidence accumulation is implemented in a straightforward manner, but the optimal boundaries are complex and nonlinear[7]. The optimal decision policy is approximated by a recurrent neural circuit that implements a nonlinear transformation of the accumulators, which simplifies the decision rule to a simple winner-takes-all rule when an accumulator reaches a single threshold. The decision rule is then simple, local, single-valued, and applies to each population independently. An interesting question is whether a similar approximation and neural implementation can be implemented with the nonlinear boundaries presented here. In principle, our model could be approximated by applying normalization (as in ref. [7]) and a nonlinear transformation of the decision variable to remove the nonlinear component of the boundary. However, it is unclear whether a recurrent network (as in ref. [7]) exists to transform the decision variables for the more complex power and oscil boundaries. If it is possible to represent the nonlinearities within a larger recurrent network, one would obtain a local, single-valued decision rule applied independently to each accumulator.

## Reproduction of other experimental findings

**Hick's law in choice RTs:** Hick's law is a benchmark result relating decision time to the number of choice alternatives. The relationship is classically log-linear in the form $\bar{RT} = a + b \log(n)$. This relationship is found for both perceptual and value-based implementations of the $n$DRM for a set cost ratio[45–47]. Our model also replicates this relationship for a number of cost ratios (Fig. 9a, colored lines). Interestingly, both the slope and intercept vary with the cost ratio $c/W$.

**SAT offset and slope:** It has been reported that increasing the number of choices $n$ results in a steeper slope and larger values for decision time versus coherency, along with a steeper slope and larger values of mean error[9]. For our model, we find that the SAT curves move away from the origin with increasing choices (Fig. 9b). Notice that the

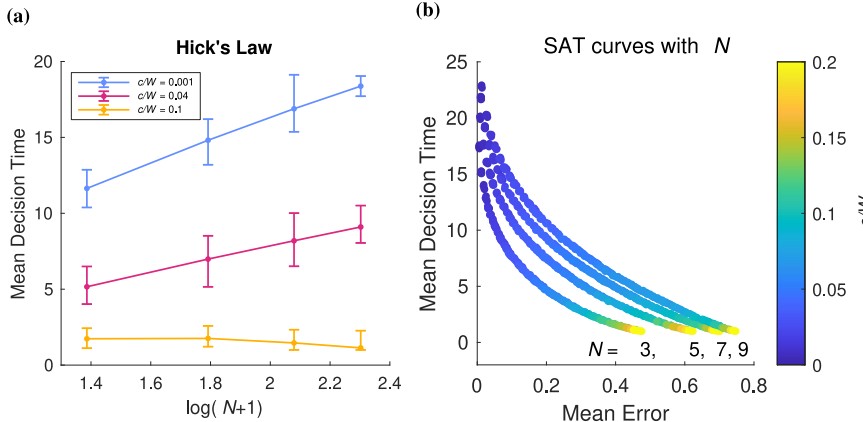

**Fig. 9 | Hick's law and SAT offset and slope. a** shows Hick's law for optimal boundaries for the examples shown in Fig. 5. Decision times are represented as mean values where error bars show the distribution (minimum/maximum) generated by the set of optimized boundaries where $n = 10{,}000$ independent decision trajectory samples. All three lines are linear and so satisfy Hick's law. **b** shows how the speed-accuracy curve varies with the number of choices $N$.

slope of the curves also decreases with the number of choices, which is not inconsistent with ref. 9.

Dynamic thresholds: Urgency signals have been observed in neural recordings from area LIP during multiple-choice tasks[8,9] and are often interpreted as implementing collapsing decision thresholds, although this is still under debate[15,16]. For multiple-choice boundaries, temporal dynamics of this type have been found for the optimized $n$DRM[7]. Our model shows that the appearance of urgency signals could, in part, be explained by the change of decision times along nonlinear decision boundaries (Figs. 7 and 8), giving rise to what looks like temporally-dynamic thresholds. However, boundaries that are nonlinear in evidence (but linear in time) do not apply to 2AFC tasks. Therefore, they cannot act as a catch-all explanation for the urgency signal since this signal has also been observed during 2AFC tasks. Additionally, our results also reproduce dynamic boundaries with increasing or mixed gradients, as found in[17], although not as a result of mixed-difficulty trials.

### Predictions

The $n$DRM makes several predictions pertaining to both behavior and neural implementation[7]. These also apply to the model presented here but stem directly from the mechanism of posterior probabilities as evidence accumulation with the consequent boundary degeneracy. Here we make a variation of these predictions and so a means of distinguishing empirically between the $n$DRM and a distinct class of models, encompassing the one here, where the integrators are normalized to represent probabilities[48,49].

Firstly, the depression of neural activity prior to evidence accumulation (the offset) increases with the number of alternatives, as found in neural recordings of area LIP[9]. We find that this can be attributed directly to the decreasing value of the priors. We, therefore, predict that mean offset is independent of modulations of the task, such as changes in reward rate or the learnt SAT. This is contrary to the predictions made by[7], which attribute the offset to a mechanism for reward maximization of the network approximation, and so predict that the offset has a dependency on reward rate, encompassing the reward values and inter-trial interval.

Secondly, the neural population activity should be near an $(n-1)$-dimensional subspace during evidence accumulation. We found that evidence accumulation takes place on an $(n-1)$-dimensional simplex with nonlinear decision boundaries. In this, we agree with[7] that the neural population activity should be constrained to an $(n-1)$-dimensional subspace during evidence accumulation, and that this could be tested with standard dimensionality-reduction techniques using multi-electrode recordings. However, there may be further subtlety in our case due to the effective degeneracy of the decision boundaries. The section "Implicit dynamics of optimal decision boundaries support both static and collapsing thresholds" showed that the decision variable can be transformed into a form that gives apparent temporal structure in the threshold (Fig. 8), dependent on the particular nonlinear boundary within the degenerate set. Our expectation is that this subtlety in the threshold dynamics may also manifest in that the accumulation manifold may appear to vary from subject to subject or trial to trial while maintaining near-optimal performance (given that any point in the accumulation manifold could also be on some decision boundary).

### Discussion

In this study, we have examined the characteristics of decision boundaries that maximize reward on a single trial of a multiple-choice decision task and found a close-to-optimal set of high-dimensional nonlinear boundaries. This differs from simple diffusion models, both qualitatively and conceptually, that are known to be optimal for two-choice tasks. We have also demonstrated that a consequence of nonlinear decision boundaries is that they can have implicit temporal dynamics that could in part contribute to apparent urgency signals in decisions with more than two choices. In addition, we proved analytically that the properties of evidence accumulation encoded by posterior probabilities are sufficient to account for a decreased offset in neural activity prior to evidence accumulation as the number of options increases, the violation of IIA and of the regularity principle, and that our model follows Hick's law, one of the most robust phenomena of perceptual decision-making[50]. However, our results do not address how nonlinear multiple-choice boundaries may be learned nor investigate the effects of choice similarity for multiple choices[51].

Our model casts light on the discussion around urgency signals and dynamic decision boundaries, along with their implications. This study demonstrates that threshold dynamics can be an implicit feature of the structure of static, high-dimensional, nonlinear decision boundaries, implemented mechanistically by either a decision boundary or a gain on evidence accumulation.

The implication is that the urgency signals reported in neural recordings of decision-making tasks with more than two alternatives, such as those recorded in area LIP[8,9], could in part be the result of a more complex decision rule. Moreover, features of the apparent urgency signal serve as an indication of the boundary shape: collapsing thresholds indicate concavity, whereas increasing thresholds (or a negative evidence gain) suggest convexity (Figs. 7 and 8). In previous work with the drift-diffusion model, increasing and mixed slopes were associated with optimality in mixed-difficulty trials[17], which gives an interpretation that some forms of convex static optimal decision rules

may generalize to mixed-difficulty trials. One way to disambiguate nonlinear decision boundaries from urgency signals is to use that this complexity only applies for multiple $n > 2$ choices, which should be apparent in neurophysiological data. However, a consensus has yet to be reached[15,16] and, as shown by ref. 7, optimal decision boundaries may be both complex and time-dependent. Thus, we predict dynamic signals on two timescales: a faster timescale for implicit dynamics from a nonlinear boundary and a slower timescale for explicit threshold dynamics that is more consistent from trial to trial.

A fundamental point is that the physical limitations of humans and animals prevent extensive reward sampling, so under such a volatile reward structure, shallow reward landscapes such as those we generated here suggest boundary parameterizations that are equally "good" in practice. A possible issue with our approach is that we have chosen sub-optimal parameterizations by limiting our cases of nonlinear boundary to curved, power-law, and oscillatory forms. However, we chose these to include the standard flat boundary case, which we find to always lie in the close-to-optimal set; moreover, our parameterizations span a broad range of nonlinear boundary shapes (Figs. 2 and 3).

In our view, the existence of a set of close-to-optimal boundaries undermines the significance of learning precise thresholds to perform multiple-choice tasks optimally. This has two implications. Firstly, emphasis is shifted onto the role of inference and normalization for multiple choices, with the shape of the boundary a secondary aspect of optimality. Secondly, this hierarchy of mechanisms for optimality, in which inference has more impact than normalization that in turn has more impact than boundary shape, has advantages for generalization and learning. There is a growing body of work demonstrating how complex behaviors can be explained through a system of inference across a range of tasks, such as the free energy principle[52]. In such systems, decision boundaries would exist implicitly as an upper bound on surprise, while action acts as a gain on the evidence much like the neural implementation described above. We propose that close-to-optimal decision boundaries may reflect the set of actions that effectively minimize surprisal. This mirrors the hierarchy of mechanisms for optimization we have found, as well as its power for generalization. Finally, the advantage for learning comes from eliminating the need for precise boundary shapes so that optimality becomes more heuristic in nature, requiring less sampling to adapt to novel tasks. The ability to generalize well to similar yet novel tasks is a generic aspect of animal behavior that is not well understood, on which the wide range of redundant near-optimal boundaries demonstrated here offers a view as to how that may be achieved.

## Methods
### Model of evidence accumulation
We consider independent $n$-choice perceptual decision tasks in which the decision-maker accumulates evidence to a boundary. To model evidence accumulation, we thus consider: (i) how the choices are represented; (ii) how evidence is sampled; and (iii) how inference is performed on the evidence to form the decision trajectories.

Choices are represented by Gaussian distributions with mean $\mu_i$ and variance $\sigma_i$, such that $i$th hypothesis $H_i$ is defined by $\mathcal{N}(\mu_i, \sigma_i)$. For more than one choice, the relative values of the means and variances determine the difficulty of the task: more overlap between distributions reduces their discriminability. We set the difficulty of all tasks using a constant difference in means $\Delta\mu$ and $\sigma = 1$.

Considering $n \geq 3$ alternatives has implications for representing the sampled variable $x(t)$. In the SPRT, $x(t)$ was drawn from one of two equivariant normal distributions $N(\mu_i, \sigma)$, $i = 0, 1$, leading to a choice symmetry that underpins the use of a single threshold for equal decision costs. However, it is impossible to maintain a symmetric overlap between three or more distributions on a line. Instead, one can only maintain symmetry by having $x(t)$ be a vector of samples with symmetric mean vectors. For simplicity, we take the means of the

equivariant normal distributions as the vertices of the $(n-1)$-simplex, with $n$ the number of hypotheses, with center at the origin and unit edge length.

Decision trajectories are generated by a series of observations on which inference is performed. Observations occur once per time step $t$ and take the form of decision evidence $x(t)$, a sample from the distribution representing the 'correct' choice. For each sample, conditional probabilities $P(x(t)|H_i)$ are calculated per hypothesis $H_i$, implicitly assuming that the number and distribution of alternatives are known to the decision-maker. The decision trajectories comprise the posterior probabilities inferred from evidence sampling, the derivation of which follows.

The SPRT is a statistical test for hypothesis testing based on the likelihood-ratio (LR) test[29]. The LR test rejects a hypothesis $H_1$ in favor of an alternative hypothesis $H_0$ if the LR is below a boundary. By the Neyman–Pearson lemma, it is the most powerful test at significance $\alpha = P(\log LR < \theta | H_1)$. In the SPRT, sequential LRs of independent samples $x(t)$ are combined iteratively into a probability ratio (PR) that is a product of the LR:

$$\log PR(t+1) = \log PR(t) + \log LR(t); \quad LR(t) = \frac{P(x(t)|H_0)}{P(x(t)|H_1)}, \quad (7)$$

which is equivalent to summing the log LRs. The SPRT continues until crossing one of two boundaries $\log PR(t) \gtrless \pm\theta$, relating to the significance of rejecting $H_0$ for $H_1$ and $H_1$ for $H_0$. (For simplicity, we consider equal-and-opposite boundaries, equivalent to equal type I and II error rates). Then SPRT is an optimal test: the optimal boundary minimizes a cost function linear in error rate and mean sample number, optimizing the trade-off between decision time and accuracy.

Sequential Bayesian inference, or Bayesian updating, is given by the Bayes rule, which updates posteriors for the two hypotheses $H_0$ and $H_1$ by combining likelihoods of the $t$th sample with priors equal to the preceding posteriors:

$$P_i(t) = P(H_i|x(1:t)) = \frac{P(x(t)|H_i)P(H_i|x(1:t-1))}{\sum_i P(x(t)|H_i)P(H_i|x(1:t-1))}, \quad i = \{0, 1\}. \quad (8)$$

This sequential update (8) is equivalent to the SPRT update (7), as verified with some simple algebra (taking logs of equation (8) and subtracting the two components, so that the denominators cancel). Then the probability ratio PR($t$) in SPRT is recognized as the posterior ratio: the SPRT decision boundaries are equivalent to posterior thresholds $P(H_i|x(1:t)) > \Theta$, where $\theta = \log(\Theta/1 - \Theta)$, and sequential Bayesian inference is identical to SPRT with the same optimality properties, suffering only from the extra computation of the marginal term in Bayes rule.

Sequential Bayesian inference can be rewritten in a form that more closely resembles a sum of evidence by taking the logarithm of equation (8):

$$\log P_i(t+1) = \log P_i(t) + \log L_i(t) - \log M(t), \quad L_i(t) = P(x(t)|H_i), \quad (9)$$

$$M(t) = \sum_i \exp(\log P_i(t) + \log L_i(t)). \quad (10)$$

For each of the $n$-choices, the accumulated evidence is the log posterior, $\log P_i(t) = \log P(H_i|x(1:t))$, with $n$ distinct evidence increments $\log L_i(t)$ and a common subtractive log-marginal term $\log M(t)$. Calculation of this log-marginal is necessary for multiple ($n > 3$) choice optimal decision-making, but can be avoided for $n = 2$ by using the SPRT as mentioned above. A similar $\log M(t)$ term in a theory of basal ganglia function has been interpreted as suppressing evidence accumulation when choices are ambiguous[53–55].

Within this framework, using that probabilities sum to unity $\sum_i P_i(t) = 1$, the evidence accumulation takes place within an $(n-1)$-

dimensional linear subspace spanning $n$ vertices $\{P_i = 1, P_{j\neq i} = 0\}$ of an $N$-dimensional unit hypercube. This is most easily seen for $N = 3$ choices, where the decision variables $\mathbf{P} = (P_1, P_2, P_3)$ vary within a triangle passing through three vertices $\{(1, 0, 0), (0, 1, 0), (0, 0, 1)\}$ of the unit cube (Fig. 1, top right panel). So rather than representing evidence accumulation trajectories as a random walk constrained to two dimensions, for multiple alternatives the random walk moves in higher dimensions (a line for two alternatives, a plane/triangle for three, and so on). For simplicity, this vector is initialized with equal prior probabilities $1/n$ and updated sequentially until one of the elements reaches a boundary, $P_i > \theta$.

Both the SPRT and two-choice sequential Bayesian inference give optimal decisions in that they give the fastest decisions for a given level of accuracy on single trials of known difficulty[5]. Formally, they optimize a cost function, the Bayes risk[25], that is linear in the mean error rate $e$ and mean decision time $T$ over many trials

$$C_{\text{risk}} = W_i \mathbb{E}(e) + c \mathbb{E}[T], \quad e = \{0, 1\}, \text{ (correct/incorrect decision)}, \quad (11)$$

with error costs $W_i > 0$ and cost of time $c$, scaling the expected error and decision time, $\mathbb{E}(e)$ and $\mathbb{E}[T]$. This cost function represents the trade-off between speed and accuracy of decision-making: slow but accurate or fast but inaccurate decisions are both costly, and so a balance must be found. For equal decision costs, $W_0 = W_1$, there is a single threshold that is free parameter, which for optimal decision-making is tuned according to the relative cost $c/W$ of accuracy and time.

An equivalent way of representing this multi-trial cost function is over single-trial rewards

$$r = \begin{cases} -W_i - cT, & \text{incorrect decision}, \\ -cT, & \text{correct decision}, \end{cases} \quad (12)$$

because the mean reward rate over many trials is $\mathbb{E}(r) = -C_{\text{risk}}$, the negative of the Bayes risk. This rewriting of the cost function emphasises that the fundamental problem in optimal decision-making is to sample decision rewards to learn the appropriate decision boundary that optimizes the reward rate.

### Decision boundary parameterizations

For multiple $n \geq 3$ alternatives, a crucial difference from 2AFCs is that decision boundaries now form hyper-planes across the decision variable space of posterior probabilities (Fig. 1). This raises the intriguing possibility that the optimal decision boundaries could instead be any two-dimensional curve for three choices and any hyper-surface for more choices. Then the boundary is not a single value, but is instead a function of the evidence, $F(\mathbf{P}(t))$.

Given the scope of possible optimal boundaries, a general parameterization is intractable. However, there are constraints that guide the construction of a family of possible parameterizations. Consider an $n$-dimensional posterior probability vector $\mathbf{P}(t) = (P_0(t), \dots, P_n(t))$ at time $t$. If any element $P_i(t) = 0$, then the boundary collapses to a boundary for $n - 1$ alternatives. For example, take $P_0$ and $P_1$ as the only non-zero probabilities, then the decision boundaries are thresholds on the line $P_0(t) + P_1(t) = 1$; these posterior thresholds $\theta$ give the edge-intersection values for higher-dimensional decision boundaries when the other probabilities $P_i, i > 2$, are non-zero. Similarly, if $P_0$, $P_1$, and $P_2$ are the only non-zero probabilities, then the decision variables lie on the 2D simplex, $P_0(t) + P_1(t) + P_2(t) = 1$, with higher-dimensional boundaries intersecting with this simplex.

Therefore, we can create a family of possible parameterizations by constraining the decision boundaries in terms of the posterior probabilities in 3D subspaces: the beliefs within all possible combinations of three choices of the $n$ alternatives available. In this way, decision boundaries are defined throughout the decision space and are easily manipulable. Furthermore, the boundaries are symmetric under

permutations $P_i \leftrightarrow P_j$ of the decision variables, which respects the assumed symmetry of the decision costs. Then we can parameterize in terms of 3D subspaces representing the outer extent of the decision space (Fig. 1) given by $P_k(t) + P_l(t) + P_m(t) = 1$, where $k \neq l \neq m$. So, the number of outer 3D subspaces within the decision space is the number of unique unordered combinations of three choices, $C_3^n = n!/3!(n-3)!$ for $n$AFCs (equaling 1, 4, 10 for $n = 3, 4, 5$ consistent with Fig. 1). Each subspace has a three-element belief vector $\mathbf{P}(t)_{\mathbf{j}} = (P_k(t), P_l(t), P_m(t))$, where $\mathbf{j}$ represents the triplets $(k, l, m)$, which collectively form the set of triplet combinations $\mathbf{j} \in \mathbf{C}_3^n$ from the integers $\{0, \dots, N-1\}$. Thus, for $n = 3$ choices, there is just one subspace $P(t)_{\{0, 1, 2\}}$; and, for $n = 4$ choices, there are four subspaces, $\{P(t)_{(0, 1, 2)}, P(t)_{(0, 1, 3)}, P(t)_{(0, 2, 3)}, P(t)_{(1, 2, 3)}\}$.

The general decision threshold parameterization is then

$$P_i(t) > F(\mathbf{P}(t); \theta, \alpha, \beta) \quad (13)$$

where

$$F = \theta \left[ 1 + \frac{\alpha}{C_3^n} \sum_{\mathbf{j} \in \mathbf{C}_3^n} f\left( \underset{\text{max}}{\Delta} \mathbf{P}(t)_{\mathbf{j}}; \beta \right) \prod_{i \in \mathbf{j}} P_i(t) \right]. \quad (14)$$

Taking the product of all components of $\mathbf{P}(t)_{\mathbf{j}}$ accomplishes two things. Firstly, it reduces threshold dimensionality by one, collapsing the decision space to the number of alternatives with non-zero belief. For example, for $n = 4$, if one alternative has a zero probability, the decision space and thresholds collapse to the $n = 3$ case (the planes shown in Fig. 1), constraining how the decision boundaries intersect with the outer faces. Secondly, it introduces the simplest form of nonlinearity, as the product of all elements $\mathbf{P}(t)_{\mathbf{j}}$ parameterizes curved boundaries.

Overall, the function $f( \underset{\text{max}}{\Delta} \mathbf{P}(t)_{\mathbf{j}}; \beta)$ modulates the curve parameterized by the product of posterior probabilities. The variable $\underset{\text{max}}{\Delta} \mathbf{P}(t)_{\mathbf{j}}$ is the maximum absolute difference between the components of the vector $\mathbf{P}(t)_{\mathbf{j}}$, which with the addition of a parameter $\beta$ allows a range of boundary shapes, such as those in equations (15–18) below. Summing these components over all subspaces generalizes (14) to $n$-choices, for which it is normalized by the number of 3D subspaces using the multiplier $1/C_3^n$.

The general parameterization, equation (14), has three parameters: $\theta$ is the intersection of the boundary with the edges in posterior probability space, because if there are only two choices with non-zero belief values, then the right-hand side of equation (14) is zero leaving $F = \theta$; $\alpha$ directly tunes the amplitude of the function $f$ or of the curve if $f = 1$; and $\beta$ is a free parameter whose purpose depends on the function $f$. Clearly, we could include additional parameters, but consider only these three for tractability.

There is no prevailing method for learning arbitrary functions with stochastic rewards, so we use the general form given by equation (14) and a range of functions $f$ over which to explore the optimality of nonlinear higher-dimensional decision boundaries. The considered functions are:

$$\text{flat}(\theta) : F_{\text{flat}}(\mathbf{P}(t); \theta), \quad f = 0, \quad (15)$$

$$\text{curve}(\theta, \alpha) : F_{\text{curve}}(\mathbf{P}(t); \theta, \alpha), \quad f = 1, \quad (16)$$

$$\text{power}(\theta, \alpha, \beta) : F_{\text{power}}(\mathbf{P}(t); \theta, \alpha, \beta), \quad f\left( \underset{\text{max}}{\Delta} \mathbf{P}(t)_{\mathbf{j}}; \beta \right) = \left( \underset{\text{max}}{\Delta} \mathbf{P}(t)_{\mathbf{j}} \right)^{\beta}, \quad (17)$$

$$\text{oscil}(\theta, \alpha, \beta) : F_{\text{oscil}}(\mathbf{P}(t); \theta, \alpha, \beta), \quad f\left( \underset{\text{max}}{\Delta} \mathbf{P}(t)_{\mathbf{j}}; \beta \right) = \cos\left( 2\pi\beta \underset{\text{max}}{\Delta} \mathbf{P}(t)_{\mathbf{j}} \right). \quad (18)$$

Examples are shown in Fig. 2. The constant-valued boundaries are dashed in all subsequent examples for comparison. Note that setting $\alpha = 0$ or $\beta = 0$ for any of (16–18) recovers the flat case (15).

Rewards are generated according to the Bayes risk represented over single trials (equation 12), when stochastic decision trajectories encounter a decision boundary. A reward landscape can be generated by systematically varying decision boundary parameters and averaging the rewards generated over a large number of simulated trajectories for each set of parameters. Averaging over many reward outcomes reduces variance producing a smooth surface and simulates the expected reward, which coincides with sampling rewards to learn the decision boundary under equation (11). So each point in the reward landscape is the mean reward obtained from a distribution of rewards for that boundary shape (typically from 100,000 samples across the parameter ranges); differing landscapes with $c/W$ discretized in 1000 steps across its range from 0 to 0.2 were then considered. Each landscape has a point of mean reward that is an absolute maximum, which we will denote by $r_{max}$; however, the landscape is very noisy even after this averaging, with many other points with similar mean rewards. Therefore, we consider a set of parameters corresponding to multiple points $x \in X$ on the landscape, with mean rewards $r_x$ and standard deviations $\sigma_x$ according to the distribution of rewards for those parameters. This set of parameters is selected such that the maximum mean reward value of the landscape falls within a $\delta\sigma_x$ of the point, so that $r_x > r_{max} - \delta\sigma_x$. Using some example reward landscapes, we found that as $\delta$ decreases, the area designated as maximum also decreases as expected. However, below $\delta = 0.02$, the area of the maximal region does not appear to change, but the points selected became more sparse, so we use this value.

### Reporting summary

Further information on research design is available in the Nature Research Reporting Summary linked to this article.

## Data availability

The data generated in this study, including the data necessary to reproduce Figs. 4–6, 8, and 9 are publicly available and have been deposited in https://doi.org/10.5523/bris.1xrlkxgze3xmu2uqc2du219wdl.

## Code availability

All code to generate and process the data and generate results is available in the Zenodo repository "DegenerateBoundaries_NatComm 22" at https://doi.org/10.5281/zenodo.6625083[56].

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

## Acknowledgements

N.F.L. and T.G. were supported by a Leverhulme Trust Research Leadership Award on "A biomimetic forebrain for robot touch" (RL-2016-039). N.F.L. was also supported by EPSRC grants on "Tactile Super-resolution Sensing" (EP/M02993X/1) and "Made Smarter Innovation" (EP/V062158/1). S.-A.B. was supported by an EPSRC DTP studentship.

## Author contributions

S.-A.B. designed and performed experiments, analysed data, and wrote the paper; T.G. edited and consulted on the paper; N.F.L. designed and performed experiments, edited the paper, and supervised the research.

## Competing interests

The authors declare no competing interests.
