## [Peer Review File · Nature Communications]

Degenerate Boundaries for Multiple-Alternative DecisionsREVIEWER COMMENTS

Reviewer #1 (Remarks to the Author):

The manuscript investigates the question of how to best commit to choices in multiple-alternative decision making. It uses a Bayesian framework for normative evidence accumulation, in which case accumulation termination is controlled by boundaries on the Bayesian posterior. The authors ask for different families of boundary shapes which member of this family maximizes the expected reward. They find that this maximum is achieved by multiple members of each family, leading to the claim that optimal boundaries are degenerate. They furthermore show that the decision-making mechanism resulting from their model replicates multiple characteristics of human multiple-alternative decision making, such as Hick's law, and violations of the regularity principle and the irrelevance of irrelevant alternatives principle.

The work closely follows previous theoretical work on multiple-alternative decision making. It is most closely related to Tajima et al. (2019) ([11] in the manuscript refers to the pre-print, but it has since been published in Nature Neuroscience) that I will in this review refer to as 'TDPP19'. The manuscript provides some novel perspectives of the consequence of complex boundaries in such decisions that make them qualitatively different from 2-AFCs. However, it also claims novelty of previously established results, and lacks precision in relating the presented work to the literature (see details below). Overall, this severely limits my enthusiasm for the presented work.

* Novelty of determining optimal decision boundaries: the authors claim that the computational principles underlying multiple-alternative decision making remains an open research question (e.g., page 2 and 3), despite the already sizable literature on this topic (e.g., multiple-alternative LCA, multiple MSPRT papers by Baum, Dragalin, Veeravalli, etc., TDPP19, to just name a few). For a specific scenario, TDPP19, for example, showed that the optimal decision boundaries are curved (Fig. 2 in TDPP19), such that "flat" boundaries (Fig. 1 in manuscript) won't be optimal. Baum & Veeravalli (1994; [12]) made a similar observation already in 1994. It might be that, due to degeneracy, a "flat" boundary performs as well as curved ones, but TDPP19 and Baum & Veeravalli (1994; [12]) suggest otherwise (see also my comments on degeneracy further below).

* Identifying optimal boundaries: from abstract and introduction it appears as if the manuscript identifies the decision boundaries that maximize overall expected reward. Later it becomes clear that the authors only consider a parametric subset of possible boundaries, and that there might be other boundary shapes outside of this subset that yield higher expected rewards. While choosing such a subset makes the work tractable, it also reduces its scope. It should be made clear from the start that "optimality" here refers to the best boundaries within the explored subset, rather than the best among all imaginable ones (which is what TDPP19 compute, and then approximate in their network model).

* Novelty of showing that decision-making takes place in an $(n-1)$ dimensional subspace: as this is a straight-forward consequence of probabilities having to sum to one, it is no surprise that this has been established before. For example, compare Figs. 1-3 in Baum & Veeravalli (1994; [12]) or Figs. 2 & 7 in TDPP19 to Fig. 1 in the manuscript. Dragalin et al. (1999a,b) also use this property.

* Degeneracy of optimal boundaries: The authors assess degeneracy by asking if expected rewards for particular parameter sets (or SATs) do not differ from the found (noisy estimate of a) maximum reward by a chosen threshold. Therefore, they don't assess degeneracy in the strict mathematical sense (which they can't from stochastic simulations), but instead ask for a range of parameters for which the expected reward is practically indistinguishable from simulations. Allowing for this slack is guaranteed to provide a range of parameters that is considered to yield comparable optimal rewards. However, in the discussion the authors go a step further and claim that finding such a range most likely implies degeneracy in the strict, mathematical sense - which isn't a step I would take. The authors furthermore note "The optimization reveals that the maxima lie along extended regions, rather than around points (figure 5).", and consider this evidence for such strict

degeneracy (as far as I understand). However, it is easy to find such extended regions even in simple functions that have different length-scales for different arguments. For example, $f(x_1, x_2) = -x_1^2 - 0.00001 * x_2^2$ would lead to a narrow range of x_1 , but a wide range of x_2 values for which $f(x_1, x_2)$ is close to its maximum, $f(0,0)=0$.

Furthermore, it is known (e.g., Baum & Veeravalli (1994; [12])) that the expected cost-to-go in the Dynamic Programming formulation of the manuscript's decision problem has a unique optimal solution (see Eq. (4) in Baum & Veeravalli (1994; [12])). Even though it doesn't rule out a non-unique optimal decision policy, it makes it less likely.

Overall, I encourage the authors to relativize their overly strong degeneracy statements, as, for example, in the title, the section heading "Optimal decision boundaries are degenerate", and related statements throughout the text. The evidence provided by the manuscript does not support such strong claims. Instead, I would suggest the authors to describe the found parameter regions as yielding close-to-optimal expected reward. This by itself is already interesting, as it means that perfect decision boundary tuning isn't really required. Pointing this out (as the authors do) doesn't require invoking degeneracy.

Side note: the chosen bound parametrization itself features degeneracy under certain circumstances. For example, the beta parameter in the power and oscillatory boundaries won't have any effect on the boundary shape if $\alpha=0$. However, this doesn't mean that the bounds themselves are degenerate.

* Impact of the considered task on the optimal boundaries: the manuscript glosses over essential qualitative differences in task setups when describing and relating their work to past literature. The SPRT was probably the first optimal stopping rule, but only applies to a very limited scenario in which the two likelihoods associated with the compared hypotheses is known. Under these circumstances, it is optimal to bound the log-likelihood ratio (or, equivalently, the posterior belief, or posterior odds), and to make decisions once a decision bound is reached.

Knowing these likelihoods implies that the decision-maker knows a-priori how hard the decision is going to be - which usually isn't the case in the real world, in which the 'strength' of decision-related evidence can vary across decisions. Under such circumstances, it has been shown in Drugowitsch et al. (2012) that the optimal decision boundary needs to depend on time. More specifically, for a bound on the posterior, this bound should collapse towards 1/2 over time.

In Drugowitsch et al. (2012) they consider a simple scenario with a single source of (perceptual) evidence. In Tajima et al. (2016) they expand this work to cases in two sources of (value-based) evidence. Despite this, they keep the setup that the evidence strength can vary across trials, leading again to decision boundaries that collapse over time. As far as I understand, if this strength would be known ahead of the decisions, the boundaries would again become time-independent. The same should apply to TDPP19, as Eq. (4) in Baum & Veeravalli (1994; [12]) also suggests.

In the manuscripts, the authors assume that the likelihoods associated with each hypothesis is known, such that the 'evidence strength' is known a-priori by the decision maker (which, in my opinion, captures few real-world scenarios). This also makes their use of time-independent decision boundaries appropriate. However, they don't make this distinction when discussing past literature, which leads to conflicting statements. For example, they state on page 2 that "Collapsing boundaries are known to be optimal for binary value-based tasks [16]", but on page 8, "(II) Contained in each of these optimal sets is the flat case (elaborated in the next section) known to be optimal for 2AFC tasks [18, 4, 27]." I urge the authors to be more precise when discussing past work, and to explicitly distinguish the aforementioned cases.

* Urgency signal: the authors provide an interesting investigation of how an urgency signal can arise in n-AFC ($n>2$) tasks from time-independent boundaries. In this context it would again be beneficial to refer to the above distinction between known and unknown evidence strength, and associated time-collapse of decision boundaries. As far as I understand, the urgency signals in Drugowitsch et al. (2012) and TDPP19 explicitly capture this time-collapse, which, due to the different task setup, isn't required in the provided manuscript. This makes the author's explanation of the urgency signal qualitatively different. What it can't do, however, is to act as a catch-all explanation for the urgency signal, as this signal has also been observed in 2-AFCs, which wouldn't be predicted by the authors' work.

* Precision on describing boundary characteristics: in the current version of the manuscript, the boundary characteristic descriptions are sometimes fairly confusing (more details below). In particular, it is neither clear from abstract nor introduction that the authors aim to investigate the consequences of decision boundaries that jointly depend on the state of multiple accumulators rather than are independent across accumulators. To make the manuscript more accessible, I suggest unpacking this distinction in more detail already in the introduction (words like "constant" and "stationary" by themselves are too ambiguous to capture the required details), as this is an essential component of the manuscript. The authors should also distinguish between time-independent and time-dependent boundaries already in the introduction, to make clear that they restrict themselves to time-independent boundaries.

* Neural representation of probabilities: the manuscript suggests that the brain represents probabilities rather than quantities from which these probabilities can be recovered (as in TDPP19 and other models). This is made explicit on, e.g., p14: "[...] that the brain performs Bayesian inference directly versus an implicit use for reward estimation." This is a much-discussed question, but there exists unfortunately little evidence for such a direct representation (and there is also no good biological reason for it). As an example to the contrary, a constant decision threshold in probabilities would imply that the probability of correct choices should be always the same at this threshold (e.g., Kira, Yang & Shadlen, 2015). However, this is not what is seen in monkeys performing the random-dot motion task, in which decision appear to be terminated at a common (neural) threshold (e.g., Roitman & Shadlen, 2002) across different motion coherences, but their choice accuracy differs at this threshold. This is just one of many examples that don't support this neural activity \leftrightarrow probability representation. I encourage the authors to discuss this critically.

Minor comments:

p2, "The coupled DDM and MSPRT have been shown to become suboptimal for more than two choices in finite time [12]." - unclear what "in finite time" means here.

p2, "All of these models assume independent, constant-valued decision boundaries for each choice accumulator, which implies that evidence accumulation and decision criteria are independent." - it is generally unclear in the introduction what it means for a boundary to be independent, and what it means that evidence accumulation and decision criteria are independent. This needs to be unpacked further to be understandable to more naive readers. Furthermore TDPP19's normative model (unlike their network model) does not assume the decision criterion to be independent across accumulators.

p3, "Constant single-valued decision rules (boundaries) are known to be optimal for 2-choice decision tasks [18]" - it isn't sufficiently clear what "constant" here means; I assume it means that the boundary does not depend on time, correct?

p3, "This is due in part to a lack of consensus on the appropriate model of inference from accumulated evidence for non-binary tasks." - I am unaware of such a lack of consensus. Could you elaborate?

p3, "We assume equal discriminability between all choices and so all choice distributions are equivariant with equidistant means (see Methods)." - I was puzzled reading this, because it is not achievable for n-AFC with $n \geq 3$ if the sampled evidence is scalar. Only in Methods does it become clear that the authors assume vector-valued evidence samples. I think this should already be discussed in the main text.

p3, Eq. (1): might be worth pointing out that this is a standard reward function that has been used in a wide range of past work (incl. TDPP19).

p4, "[...], we can consider that the SPRT can be constructed [...]" - the SPRT hasn't been introduced in detail in this point, such that this statement will remain cryptic to readers unfamiliar with the SPRT.

p4, Eq. (2) is generally referred to as log-odds or log-likelihood ratio (e.g., Gold & Shadlen (2007)); would be good to mention for readers familiar with these concepts.

p4, "Now, the key point is that sequential Bayesian inference applies to arbitrary number of choices, and so holds for general nAFC decision making." - at this point it might be worth citing [12], which already made this point. Furthermore, they provide figures very similar to Fig. 1a.

p6, "So for $n > 2$ choices, boundaries can have spatial dependence representing probability distributions over choices (figures 1a, 1b)." - I did not understand this sentence. Please clarify.

p6, "Our interest is in whether these or any other spatial function could give improved performance over the single-valued case." - might be worth pointing out here (or elsewhere) that TDPP19 have shown that curved decision boundaries are optimal, but that this does not exclude the possibility that other decision boundaries perform as well as curved ones (see above comment about uniqueness of optimal cost-to-go, but potential non-uniqueness of optimal policy).

Fig 4: I would have liked to also know which of the different functional forms of the boundary achieves better overall performance. From Figure 4 it appears that the "curve" boundaries achieve a lower mean error for the same mean decision time than other boundary functions, and so should be overall better. Is this intuition correct?

p10, "Degenerate optimal decision boundaries are qualitatively different" - different from what?

p10, "If the degenerate optimal decision boundaries are qualitatively similar for given costs, then learning the correct boundary shape would be vital for optimal decision making." - I would have assumed the opposite. I find the whole paragraph unclear and suggest revising it.

Fig 6: the caption mentions that "Panels (a-c), show the decision time distribution over the belief space, [...]" but it is unclear how they do so. I would have expected decisions to only happen at the decision boundaries, such that the decision time distributions should be restricted to these decision boundaries. Instead, they seem to cover the whole belief simplex. Why is that?

p11, "[...], contrasting with 2AFCs where decision boundaries are zero-dimensional and so limited to a single mean error and mean decision time." - this is unclear as decision boundaries extend over time, and (by definition) yield different decision times at different boundary hitting times. You seem to allow fixing the belief (or log-odds) for the chosen option while allowing those associated with the unchosen option to vary, but don't allow fixing the decision time; should be made more explicit.

Fig 7: Do I understand correctly that thresholds that increase over time arise in scenarios in which later choices are more likely correct than earlier choices? If yes, this could be assessed in experiments from the time-dependent accuracy plot (e.g., Kira, Yang & Shadlen (2015)), and would be worth pointing out.

Fig 7, caption: "The first three rows [...]" - there are only three rows; what do other rows show? Unclear.

p13, "[...] the models have a different reward structure and optimization process, [...]" - as far as I can tell, the models have similar reward structure (i.e., Eq. (1) is also used by TDPP19), but different evidence structure (i.e., latent states are drawn from a Gaussian rather than from a discrete set).

p14, "Network model" - this section provides a very high-level discussion on the similarities and differences of the manuscript's decision model to TDPP19's network model. However, I am not sure I understand much of this section, and encourage the authors to revise it.

p19, "The majority of multi-hypothesis studies use multiple competing SPRTs (known as MSPRT) [48, 44]." - this might be the case in the neuroscience literature (and even there it isn't that frequent, to my

knowledge), but in the one that actually introduced the MSPRT (e.g., work by Veeravalli and colleagues), the works usually don't do this.

p19, Eq. (8): the notation is ambiguous, as $x(t)$ is used to, on one hand, denote the evidence sample at time t , and, on the other hand, denote all evidence samples until time t . Please dis-ambiguate.

p22, "[...] within a small $\delta = 0.02 \sigma_i$ standard deviation from this point" - the information that the "acceptance region" around the maximum is determined by the standard deviation should go into the main text. Furthermore, given that it is means that are compared, why not use the standard error of the mean to define the threshold? Admittedly, for a fixed sample size, those two definitions will be equivalent, but, conceptually, it would make, in my opinion, more sense.

References, [11] - TDPP19 was published in Nature Neuroscience.

References, [46] - first author's name is spelled incorrectly.

Reviewer #2 (Remarks to the Author):

This paper describes a surprising theoretic result relating to optimal choices between multiple options. Theory of decision making has established that the optimal decision procedures involve integrating sensory information until the evidence for one of the options reaches a specific threshold referred to as a decision boundary. Over the last 20 years, much research has focussed on identifying the shape of the optimal decision boundary for different tasks involving choice between two options, and comparing the optimal shape with experimental data. This manuscript shows that for the choice between more than two options the optimal boundary may not be unique, but for each analysed choice problem there exist multiple boundaries giving indistinguishable reward. To my surprise, this happens even for the simplest possible scenario of choice with known difficulty, analysed in this paper.

This paper has an interesting implication that for more than two options it is not worth asking what the optimal decision boundary is, because it may not be unique. Hence the manuscript is a useful contribution, as it will inform further research on its topic. However, several points in the manuscript require further clarification.

Major comments:

1. As the Authors admit, the results of the paper are numeric rather than analytic, and hence they do not reveal if multiple boundaries give the same or just very similar reward. Nevertheless more numeric investigation of this question would be useful. For example in Figure 5a it is unclear if multiple sets of parameters exist OR there is just more sensitivity to θ than to α . To shed more light on this question, it would be good to plot R for $\alpha = \{-20, -10, 0, 10, 20\}$ for optimal θ s for a given α from Figure 5a with 100 times more iterations than in Figure 5a (which would be feasible as the plot would just include 5 points) and display also error bars showing standard error.
2. Figure 6 is an important figure in this paper, because it aims at explaining the reason for the existence of multiple equally good boundaries, but at the moment it is difficult to read. The caption says: "Examples of qualitatively different degenerate decision boundaries, overlaid on the mean decision time and mean error" – I do not see the mean error on this figure. Have the Authors forgotten to include the panels with error? If so, please add them, or otherwise clarify how to read out the error from this figure. Also the colour scale in Figure 6 is difficult to read. In the text the Authors made an interesting observation: "decision times vary even along the flat boundaries (figure 6, yellow lines)", but this is very difficult to see. Can the figure be modified to make this surprising effect easier to see?
3. This manuscript sheds light on questions which have been puzzling me related to existing tests for decision making between multiple options. It has been already shown by Dragalia et al. (1999 IEEE Transactions on Information Theory) that there exist two tests MSPRT_a and MSPRT_b which both are asymptotically optimal in a limit of error going to 0. Since these tests are well known and often studied, it would be helpful to analyse them in the framework of this paper. I have been always wondering: Which one of them is really better for larger error rates and why do they become

equivalent in the limit. I feel that the methodology developed in the manuscript can help answer these questions. First, it would be interesting to explicitly compare the shape of the boundaries of the two tests and visualize them on the “triangle plots” (like Figure 6). MSPRTa corresponds to “flat boundaries”, and it would be interesting to see the shape for MSPRTb. Second, it would be nice to compare in simulations the rewards for the two tests. Third, does the asymptotic optimality relate to the fact that in the limit of error $\rightarrow 0$ the bounds become a point in the “triangle plots” for both tests, and therefore the tests become equivalent?

Minor comments:

1. Figure 4: It is not clear if all parameters have been optimized, or just theta. If the latter, please give the values of other parameters.

2. In Section “Implicit dynamics of optimal decision boundaries support both static and collapsing thresholds” the Authors wrote: “the boundary beliefs on choice selection, as generated by Monte-Carlo decision trajectories, are transformed to a log-likelihood representation (equation (2))”. It is not clear to me why this transformation is performed – could you please justify? (Equation 2 was introduced in the context of 2 alternatives, while this section is about 3 alternatives, so it is not clear how it is relevant).

3. In Discussion the Authors say: “The implication is that the urgency signals reported in neural recordings of decision making tasks, such as those recorded in area LIP [7, 8], could in fact be the hallmark of a more complex decision rule, and that the signal as a distinct phenomenon may be misleading as to the true mechanism.” I feel this statement is too strong for 2 reasons: First, the urgency signals have also been observed for 2 alternatives (see Figure 4a in Churchland et al. 2006 Nature Neuroscience), while the theory presented by the Authors does not predict any varying boundaries for two options. Second, the experiments investigating urgency signals (e.g. by Churchland et al. 2006 Nature Neuroscience) involved trials with mixed difficulty, while the manuscript considers constant difficulty. As Authors write later, it has been demonstrated by Drugowitsch et al (2012 J Neurosci) that when evidence varies in difficulty, as in these experimental studies, then the urgency signals may indeed improve reward rate. Due to the differences in the assumptions of the manuscript and the experimental studies investigating urgency signals the statement cited above needs to be tempered.

4. In section “Predictions”, the Authors say that their model predicts that initial activity should depend on the number of alternative choices. I feel it needs to be clarified that this prediction is not unique to the model described in the manuscript, but applies to a more general class of models in which the activities of integrators are normalized to represent the probabilities (e.g. Ditterich 2010 Frontiers Neurosci.; Bogacz & Larsen 2011 Neural Computation. The Authors may find interesting a comparison of this prediction with experimental data in a book chapter by Bogacz (2009, Optimal decision-making theories. In Handbook of reward and decision making, pp. 373-397. Academic Press).

5. On page 21: “For n choices, the decision space has C^n_3 faces” – I do not understand what it means, and had problems with following the text below. Please clarify.

Response to comments on: ‘Degenerate Boundaries for Multiple-Alternative Decisions’

Submitted to Nature Communications (NCOMMS-20-24534-T)

Author response to Reviewer #1

The manuscript investigates the question of how to best commit to choices in multiple-alternative decision making. It uses a Bayesian framework for normative evidence accumulation, in which case accumulation termination is controlled by boundaries on the Bayesian posterior. The authors ask for different families of boundary shapes which member of this family maximizes the expected reward. They find that this maximum is achieved by multiple members of each family, leading to the claim that optimal boundaries are degenerate. They furthermore show that the decision-making mechanism resulting from their model replicates multiple characteristics of human multiple-alternative decision making, such as Hick's law, and violations of the regularity principle and the irrelevance of irrelevant alternatives principle.

The work closely follows previous theoretical work on multiple-alternative decision making. It is most closely related to Tajima et al. (2019) ([11] in the manuscript refers to the pre-print, but it has since been published in Nature Neuroscience) that I will in this review refer to as 'TDPP19'. The manuscript provides some novel perspectives of the consequence of complex boundaries in such decisions that make them qualitatively different from 2-AFCs. However, it also claims novelty of previously established results, and lacks precision in relating the presented work to the literature (see details below). Overall, this severely limits my enthusiasm for the presented work.

We thank the reviewer for their very thorough and expert review of this manuscript, and for the time they have clearly spent. These comments have led us to make some major changes to the manuscript that we describe below.

* Novelty of determining optimal decision boundaries: the authors claim that the computational principles underlying multiple-alternative decision making remains an open research question (e.g., page 2 and 3), despite the already sizable literature on this topic (e.g., multiple-alternative LCA, multiple MSPRT papers by Baum, Dragalin, Veeravalli, etc., TDPP19, to just name a few). For a specific scenario, TDPP19, for example, showed that the optimal decision boundaries are curved (Fig. 2 in TDPP19), such that "flat" boundaries (Fig. 1 in manuscript) won't be optimal. Baum & Veeravalli (1994; [12]) made a similar observation already in 1994. It might be that, due to degeneracy, a "flat" boundary performs as well as curved ones, but TDPP19 and Baum & Veeravalli (1994; [12]) suggest otherwise (see also my comments on degeneracy further below).

We did not mean to give the impression that little work had been done on multiple-alternative decision making, but rather the form of the optimal boundaries remains an unresolved question of significant interest. We have made substantial edits to the introduction to cover related work.

There are other comments about degeneracy that we will respond to below.

* Identifying optimal boundaries: from abstract and introduction it appears as if the manuscript identifies the decision boundaries that maximize overall expected reward. Later it becomes clear that the authors only consider a parametric subset of possible boundaries, and that there might be other boundary shapes outside of this subset that yield higher expected rewards. While choosing such a subset makes the work tractable, it also reduces its scope. It should be made clear from the start that "optimality" here refers to the best boundaries within the explored

subset, rather than the best among all imaginable ones (which is what TDPP19 compute, and then approximate in their network model).

We agree and have added text to the Abstract and Introduction to make this point clear to the reader. The section on 'Multi-dimensional decision boundaries can be complex' also clarifies this further.

* Novelty of showing that decision-making takes place in an (n-1) dimensional subspace: as this is a straight-forward consequence of probabilities having to sum to one, it is no surprise that this has been established before. For example, compare Figs. 1-3 in Baum & Veeravalli (1994; [12]) or Figs. 2 & 7 in TDPP19 to Fig. 1 in the manuscript. Dragalin et al. (1999a,b) also use this property.

We agree and have included text in the section 'Multi-alternative decision making as a particle diffusing in n dimensions'

'In this section, we show that n-alternative decision making can be viewed as a diffusion process in an (n-1) dimensional subspace of the belief space. This is a perspective that has previously been established (for example, see [12, 11]), but we cover this material here to help the reader build intuition and to detail the implications for multi-alternative decisions.'

Please also note we have also removed discussion 'point 1' that discussed this as a novel point.

* Degeneracy of optimal boundaries: The authors assess degeneracy by asking if expected rewards for particular parameter sets (or SATs) do not differ from the found (noisy estimate of a) maximum reward by a chosen threshold. Therefore, they don't assess degeneracy in the strict mathematical sense (which they can't from stochastic simulations), but instead ask for a range of parameters for which the expected reward is practically indistinguishable from simulations. Allowing for this slack is guaranteed to provide a range of parameters that is considered to yield comparable optimal rewards. However, in the discussion the authors go a step further and claim that finding such a range most likely implies degeneracy in the strict, mathematical sense - which isn't a step I would take. The authors furthermore note "The optimization reveals that the maxima lie along extended regions, rather than around points (figure 5).", and consider this evidence for such strict degeneracy (as far as I understand). However, it is easy to find such extended regions even in simple functions that have different length-scales for different arguments. For example, $f(x_1, x_2) = -x_1^2 - 0.00001 * x_2^2$ would lead to a narrow range of x_1 , but a wide range of x_2 values for which $f(x_1, x_2)$ is close to its maximum, $f(0,0)=0$.

Furthermore, it is known (e.g., Baum & Veeravalli (1994; [12])) that the expected cost-to-go in the Dynamic Programming formulation of the manuscript's decision problem has a unique optimal solution (see Eq. (4) in Baum & Veeravalli (1994; [12])). Even though it doesn't rule out a non-unique optimal decision policy, it makes it less likely.

Overall, I encourage the authors to relativize their overly strong degeneracy statements, as, for example, in the title, the section heading "Optimal decision boundaries are degenerate", and related statements throughout the text. The evidence provided by the manuscript does not support such strong claims. Instead, I would suggest the authors to describe the found parameter regions as yielding close-to-optimal expected reward. This by itself is already interesting, as it means that perfect decision boundary tuning isn't really required. Pointing this out (as the authors do) doesn't require invoking degeneracy.

This is an important point that we have taken completely on board. We were aware of this when writing the manuscript but stated the results too strongly in giving the impression of

claiming optimality in the strict mathematical sense. We have phrased the claim instead in terms of an 'effective degeneracy' and weakened the statements relating to optimality (including e.g. changing the paper's title).

Because there are major revisions relating to this point throughout the paper, it is difficult to list them all here, but note in particular the section 'A degenerate set of decision boundaries yield close-to-optimal expected reward', where most of the text is new and has been framed around this point.

Side note: the chosen bound parametrization itself features degeneracy under certain circumstances. For example, the beta parameter in the power and oscillatory boundaries won't have any effect on the boundary shape if $\alpha=0$. However, this doesn't mean that the bounds themselves are degenerate.

The parameterisations were designed to include the flat boundary $\alpha=0$ case, so will necessarily become over-defined at that point, which we have clarified with 'We have chosen these parameterizations so that if $\beta=0$ we recover the curve parameterization, and if $\alpha=0$ we recover the flat parameterization.'

We have been careful to show that the set of reward-maximising decision boundaries are meaningfully different, for example in the section: 'Effectively-degenerate optimal decision boundaries are qualitatively varied', most of which concerns this point.

* Impact of the considered task on the optimal boundaries: the manuscript glosses over essential qualitative differences in task setups when describing and relating their work to past literature. The SPRT was probably the first optimal stopping rule, but only applies to a very limited scenario in which the two likelihoods associated with the compared hypotheses is known. Under these circumstances, it is optimal to bound the log-likelihood ratio (or, equivalently, the posterior belief, or posterior odds), and to make decisions once a decision bound is reached.

Knowing these likelihoods implies that the decision-maker knows a-priori how hard the decision is going to be - which usually isn't the case in the real world, in which the 'strength' of decision-related evidence can vary across decisions. Under such circumstances, it has been shown in Drugowitsch et al. (2012) that the optimal decision boundary needs to depend on time. More specifically, for a bound on the posterior, this bound should collapse towards $1/2$ over time.

In Drugowitsch et al. (2012) they consider a simple scenario with a single source of (perceptual) evidence. In Tajima et al. (2016) they expand this work to cases in two sources of (value-based) evidence. Despite this, they keep the setup that the evidence strength can vary across trials, leading again to decision boundaries that collapse over time. As far as I understand, if this strength would be known ahead of the decisions, the boundaries would again become time-independent. The same should apply to TDPP19, as Eq. (4) in Baum & Veeravalli (1994; [12]) also suggests.

In the manuscripts, the authors assume that the likelihoods associated with each hypothesis is known, such that the 'evidence strength' is known a-priori by the decision maker (which, in my opinion, captures few real-world scenarios). This also makes their use of time-independent decision boundaries appropriate. However, they don't make this distinction when discussing past literature, which leads to conflicting statements. For example, they state on page 2 that "Collapsing boundaries are known to be optimal for binary value-based tasks [16]", but on page 8, "(II) Contained in each of these optimal sets is the flat case (elaborated in the next

section) known to be optimal for 2AFC tasks [18, 4, 27]." I urge the authors to be more precise when discussing past work, and to explicitly distinguish the aforementioned cases.

We have been careful when amending the manuscript so that task specifics are paired with any discussion of optimal boundaries when we talk about previous findings and our own findings. Again, this is a point that threads through the paper, so it is difficult to list all cases here, but hopefully the manuscript is now satisfactory in this regard.

* Urgency signal: the authors provide an interesting investigation of how an urgency signal can arise in n-AFC ($n > 2$) tasks from time-independent boundaries. In this context it would again be beneficial to refer to the above distinction between known and unknown evidence strength, and associated time-collapse of decision boundaries. As far as I understand, the urgency signals in Drugowitsch et al. (2012) and TDPP19 explicitly capture this time-collapse, which, due to the different task setup, isn't required in the provided manuscript. This makes the author's explanation of the urgency signal qualitatively different. What it can't do, however, is to act as a catch-all explanation for the urgency signal, as this signal has also been observed in 2-AFCs, which wouldn't be predicted by the authors' work.

Thank you for appreciating this point. We have made substantive changes to the section 'Implicit dynamics of optimal decision boundaries support both static and collapsing threshold' to capture these subtleties

We have also excluded the 2-AFC case for the reasons you have said e.g.

Introduction: 'Our results suggest a previously unconsidered component to the origin of urgency signals in decision tasks with greater than two choices,...'

Discussion: 'The implication is that the urgency signals reported in neural recordings of decision-making tasks with more than two alternatives,...'

* Precision on describing boundary characteristics: in the current version of the manuscript, the boundary characteristic descriptions are sometimes fairly confusing (more details below). In particular, it is neither clear from abstract nor introduction that the authors aim to investigate the consequences of decision boundaries that jointly depend on the state of multiple accumulators rather than are independent across accumulators. To make the manuscript more accessible, I suggest unpacking this distinction in more detail already in the introduction (words like "constant" and "stationary" by themselves are too ambiguous to capture the required details), as this is an essential component of the manuscript. The authors should also distinguish between time-independent and time-dependent boundaries already in the introduction, to make clear that they restrict themselves to time-independent boundaries.

Thank you for bringing this to our attention. We have made several changes to the manuscript to clarify boundary characteristics, including:

Abstract: 'We show that, within a parametric subset of time-independent boundaries, the optimal decision boundaries comprise a degenerate set of complex structures that jointly depend on the state of multiple accumulators, and speed-accuracy trade-offs, contrary to current 2-choice results.'

Introduction: Extensive re-phrasing including, 'Almost all of these models assume independent, constant-valued decision boundaries for each choice accumulator (except [11]), meaning decision boundaries that are independent across accumulators and are not dynamic, i.e. not a function of either time or the state(s) of other accumulators; this implies that evidence

accumulation and decision criteria are independent.’ and several other rewritten sentences. Hopefully the manuscript is now satisfactory in this regard.

* Neural representation of probabilities: the manuscript suggests that the brain represents probabilities rather than quantities from which these probabilities can be recovered (as in TDPP19 and other models). This is made explicit on, e.g., p14: "[...] that the brain performs Bayesian inference directly versus an implicit use for reward estimation." This is a much-discussed question, but there exists unfortunately little evidence for such a direct representation (and there is also no good biological reason for it). As an example to the contrary, a constant decision threshold in probabilities would imply that the probability of correct choices should be always the same at this threshold (e.g., Kira, Yang & Shadlen, 2015). However, this is not what is seen in monkeys performing the random-dot motion task, in which decision appear to be terminated at a common (neural) threshold (e.g., Roitman & Shadlen, 2002) across different motion coherences, but their choice accuracy differs at this threshold. This is just one of many examples that don't support this neural activity <-> probability representation. I encourage the authors to discuss this critically.

This is a good point. We have responded with two main changes:

First, the "Bayesian Inference as evidence accumulation" subsection now makes clear the presented model is not intended to have biophysical realism but is to explore the effect of non-linear boundaries on multi-alternative decision-making. We also make the point that current evidence suggests that the brain represents evidence through quantities from which posterior beliefs could be inferred rather than representing those beliefs directly as probabilities.

Second, in the subsection ‘Comparison with the nDRM’ we have framed our coverage with: ‘This is because there is actually little supporting evidence for the brain representing posterior beliefs directly as probabilities; instead, studies point towards the brain employing indirect representations from which beliefs can be inferred [55, 56]. However, we describe below how either view of evidence representation is compatible for two of our main results.’

Minor comments:

* p2, "The coupled DDM and MSPRT have been shown to become suboptimal for more than two choices in finite time [12]." - unclear what "in finite time" means here.

We have clarified this by rephrasing (introduction, first paragraph):

‘However, the MSPRT has only been shown to be asymptotically optimal in the limit of vanishing decision errors’

* p2, "All of these models assume independent, constant-valued decision boundaries for each choice accumulator, which implies that evidence accumulation and decision criteria are independent." - it is generally unclear in the introduction what it means for a boundary to be independent, and what it means that evidence accumulation and decision criteria are independent. This needs to be unpacked further to be understandable to more naive readers. Furthermore TDPP19's normative model (unlike their network model) does not assume the decision criterion to be independent across accumulators.

We agree and have unpacked the meaning of independence towards the end of the first paragraph of the introduction:

“Here we explore this aspect, aiming to investigate the consequences of decision boundaries that jointly depend on the state of multiple accumulators rather than are independent.”

And also state:

“Almost all of these models assume independent, constant-valued decision boundaries for each choice accumulator (except [11]).”

* p3, "Constant single-valued decision rules (boundaries) are known to be optimal for 2-choice decision tasks [18]" - it isn't sufficiently clear what "constant" here means; I assume it means that the boundary does not depend on time, correct?

Yes, this text has been amended to (Problem setup and aim, page 3):

“Constant single-valued decision rules (boundaries), where the boundary does not depend on time, are known to be optimal for 2-choice decision tasks”.

* p3, "This is due in part to a lack of consensus on the appropriate model of inference from accumulated evidence for non-binary tasks." - I am unaware of such a lack of consensus. Could you elaborate?

On reflection we felt this sentence just confused matters and was not needed, so it has been removed from the manuscript.

* p3, "We assume equal discriminability between all choices and so all choice distributions are equivariant with equidistant means (see Methods)." - I was puzzled reading this, because it is not achievable for n-AFC with $n \geq 3$ if the sampled evidence is scalar. Only in Methods does it become clear that the authors assume vector-valued evidence samples. I think this should already be discussed in the main text.

We agree; the text has been amended to “We assume equal discriminability between all choices and so all choice distributions are equivariant with equidistant means, this is achieved by using vector-valued evidence samples (see Methods).” in the first paragraph of the section “Results: Problem setup and aim”.

* p3, Eq. (1): might be worth pointing out that this is a standard reward function that has been used in a wide range of past work (incl. TDPP19).

Yes, we have added the following text to (page 3):

“This is a standard reward function used in a wide range of past work, for example [18, 11]”

* p4, "[...], we can consider that the SPRT can be deconstructed [...]" - the SPRT hasn't been introduced in detail in this point, such that this statement will remain cryptic to readers unfamiliar with the SPRT.

We have removed the reference to the SPRT in the Introduction and replaced with more general description of the approach we take:

"We first formulate an n-dimensional Bayesian framework for decision-making between n choices and then compose a set of complex nonlinear boundary parameterizations that when optimized reveal a degenerate set of reward-maximizing decision strategies."

Amended the sentence you reference on p.4 and have also directed the reader to the Methods section for details on the SPRT earlier in the section.

* p4, Eq. (2) is generally referred to as log-odds or log-likelihood ratio (e.g., Gold & Shadlen (2007)); would be good to mention for readers familiar with these concepts.

We agree and have cited their work.

* p4, "Now, the key point is that sequential Bayesian inference applies to arbitrary number of choices, and so holds for general nAFC decision making." - at this point it might be worth citing [12], which already made this point. Furthermore, they provide figures very similar to Fig. 1a.

Thanks, we have included this citation.

* p6, "So for $n > 2$ choices, boundaries can have spatial dependence representing probability distributions over choices (figures 1a, 1b)." - I did not understand this sentence. Please clarify.

The sentence has been clarified to (page 6):

“So for $n > 2$ choices, boundaries can have spatial dependence with respect to the visualisation of decision space shown in figures 1a and 1b.”

* p6, "Our interest is in whether these or any other spatial function could give improved performance over the single-valued case." - might be worth pointing out here (or elsewhere) that TDPP19 have shown that curved decision boundaries are optimal, but that this does not exclude the possibility that other decision boundaries perform as well as curved ones (see above comment about uniqueness of optimal cost-to-go, but potential non-uniqueness of optimal policy).

We agree, and have amended the text to (page X)

“It has been shown in [11], for 3AF, curved decision boundaries perform optimally, however this does not exclude the possibility that other decision boundaries are likewise optimal. Our interest is therefore in whether these or any other spatial function give improved performance over the single-valued case, and so perform comparably to the boundaries found in [11].”

* Fig 4: I would have liked to also know which of the different functional forms of the boundary achieves better overall performance. From Figure 4 it appears that the "curve" boundaries achieve a lower mean error for the same mean decision time than other boundary functions, and so should be overall better. Is this intuition correct?

This may be the case, although it is unclear because the results are so close. We have mentioned this in the section 'Complex decision boundaries are consistent with the speed-accuracy curve': The curve parameterization appears, marginally, to yield the best SAT curve, which is consistent with previous work since this is the shape of the optimal boundary found for the 3AFC case in [12, 11]. However, the small differences between the SAT curves indicate that a wide range of boundary characteristics are still capable of generating near optimal decisions. Moreover, each value of W/c has many overlapping points on the curve, signifying that the SAT can be satisfied by multiple optimal boundaries even within the same parameterization.

* p10, "Degenerate optimal decision boundaries are qualitatively different" - different from what?

The issue was our phrasing – we meant ‘varied’:

“Effectively-degenerate optimal decision boundaries can vary qualitatively.”

* p10, "If the degenerate optimal decision boundaries are qualitatively similar for given costs, then learning the correct boundary shape would be vital for optimal decision making." - I would have assumed the opposite. I find the whole paragraph unclear and suggest revising it.

We have significantly revised the section (p.11-13) to clarify the objectives and arguments that develop the next section.

* Fig 6: the caption mentions that "Panels (a-c), show the decision time distribution over the belief space, [...]" but it is unclear how they do so. I would have expected decisions to only happen at the decision boundaries, such that the decision time distributions should be restricted to these decision boundaries. Instead, they seem to cover the whole belief simplex. Why is that?

This is a good point; we have provided clarification on page 13 (now caption for Figure 7): "Panels (a-c), show a map of the mean decision time over the belief space, with low decision times at the centre increasing towards the edges. Decisions, and hence decision times, only manifest at decision boundaries, so to visualise the variation over the whole belief simplex, we generated an approximation of the mean decision time distribution using incrementally increasing flat decision boundaries."

Please note that although we use an approximation using flat boundaries, the phenomenon is real as shown in fig 7.

* p11, "[...], contrasting with 2AFCs where decision boundaries are zero-dimensional and so limited to a single mean error and mean decision time." - this is unclear as decision boundaries extend over time, and (by definition) yield different decision times at different boundary hitting times. You seem to allow fixing the belief (or log-odds) for the chosen option while allowing those associated with the unchosen option to vary, but don't allow fixing the decision time; should be made more explicit.

We have amended the text as follows (pages 11-12):

“An aspect of multiple-choice decision boundaries is that every point on a boundary has an associated error and decision time distribution (see figure 7 for a map on the diffusing plane of the mean decision time for flat boundaries of different magnitudes). In contrast, 2AFC decision boundaries are zero-dimensional (points on the line connecting two choice apexes, e.g. the line connecting choice H1 and H2 in figure 1a) and so each choice boundary is a single point with an error and decision time distribution -- there is no boundary shape over which the error and decision time distribution can vary. Spatially dependent error and decision time distributions (varying along the decision boundary) is a consequence of having an n-dimensional belief vector that can vary over a decision boundary embedded as a curve or surface in the higher-dimensional belief space, this phenomenon and comparison to 2AFC boundaries within this framework therefore holds even if time-dependent thresholds are considered for both cases.”

* Fig 7: Do I understand correctly that thresholds that increase over time arise in scenarios in which later choices are more likely correct than earlier choices? If yes, this could be assessed in experiments from the time-dependent accuracy plot (e.g., Kira, Yang & Shadlen (2015)), and would be worth pointing out.

This is a good point that we have added to the paper (p. 14) and included a citation for your suggested reference, which as you say contains some interesting plots of the time-dependent accuracy, including perhaps an example of increasing decision threshold from Monkey J, resembling one our Increasing thresholds from the "power" parameterization (Figure 8b).

* Fig 7, caption: "The first three rows [...]" - there are only three rows; what do other rows show? Unclear.

New Fig 7 caption:

"The rows show examples of implicit dynamics categorised as increasing, decreasing (collapsing), and flat (static), colors indicate the cost ratio W/c for which the thresholds are optimized."

* p13, "[...] the models have a different reward structure and optimization process, [...]" - as far as I can tell, the models have similar reward structure (i.e., Eq. (1) is also used by TDPP19), but different evidence structure (i.e., latent states are drawn from a Gaussian rather than from a discrete set).

We have removed the incorrect statement about the reward structure and amended the sentence so it now reads: "[...] the models have a different evidence structure, trial structure and optimization process, [...]". We have added a reference to the difference in how decision trials are set up because TDPP19 assumes "the decision-maker is free to choose at any time within each trial and proceeds through a long sequence of trials within a fixed time period", whereas we assume that trials are independent with no overall time constraint.

* p14, "Network models" - this section provides a very high-level discussion on the similarities and differences of the manuscript's decision model to TDPP19's network model. However, I am not sure I understand much of this section, and encourage the authors to revise it.

We agree and have completely rewritten this subsection.

* p19, "The majority of multi-hypothesis studies use multiple competing SPRTs (known as MSPRT) [48, 44]." - this might be the case in the neuroscience literature (and even there it isn't that frequent, to my knowledge), but in the one that actually introduced the MSPRT (e.g., work by Veeravalli and colleagues), the works usually don't do this.

On reflection, this short paragraph is not necessary for the development of the methods section and so we have removed it.

* p19, Eq. (8): the notation is ambiguous, as $x(t)$ is used to, on one hand, denote the evidence sample at time t , and, on the other hand, denote all evidence samples until time t . Please disambiguate.

Thank you for bringing this to our attention: we have used $x(t)$ to denote evidence sample at time t , and $x(1:t)$ to denote all evidence samples until time t .

* p22, "[...] within a small $\delta = 0.02 \sigma_i$ standard deviation from this point" - the information that the "acceptance region" around the maximum is determined by the standard deviation should go into the main text. Furthermore, given that it is means that are compared, why not use the standard error of the mean to define the threshold? Admittedly, for a fixed

sample size, those two definitions will be equivalent, but, conceptually, it would make, in my opinion, more sense.

We have added more details about this in the main text:

‘Optimal parameters were extracted by taking all points with mean reward for which $r > r_{\text{max}} - \delta \sigma$, where r_{max} is the maximum mean reward of the noisy landscape, σ is the spread of rewards, and $\delta=0.02$ is a small parameter (see Methods for details).’

In addition, we have expanded these details both in the main text and in the methods section.

As for the standard error of the mean, as you say because we used a fixed size, using the standard deviation and standard error of the mean are in effect equivalent. To our minds, it helps clarify the arguments to have δ as a small parameter (noting some more details have been added on this in the methods), which would fit with using the standard deviation. We did consider putting in both interpretations but felt this would complicate matters.

* References, [11] - TDPP19 was published in Nature Neuroscience.

We have updated this reference, thank you .

* References, [46] - first author's name is spelled incorrectly.

We have gone back to the original reference and could not find an issue with V.P. Draglia, A.G. Tartakovsky, V.V. Veeravalli

e.g. here: <https://ieeexplore.ieee.org/document/796383>

Please let us know if there is something we are missing. We have noticed some inconsistencies in how this paper is cited in the literature but cannot find an issue with our citation.

Author response to Reviewer #2

This paper describes a surprising theoretic result relating to optimal choices between multiple options. Theory of decision making has established that the optimal decision procedures involve integrating sensory information until the evidence for one of the options reaches a specific threshold referred to as a decision boundary. Over the last 20 years, much research has focussed on identifying the shape of the optimal decision boundary for different tasks involving choice between two options, and comparing the optimal shape with experimental data. This manuscript shows that for the choice between more than two options the optimal boundary may not be unique, but for each analysed choice problem there exist multiple boundaries giving indistinguishable reward. To my surprise, this happens even for the simplest possible scenario of choice with known difficulty, analysed in this paper.

This paper has an interesting implication that for more than two options it is not worth asking what the optimal decision boundary is, because it may not be unique. Hence the manuscript is a useful contribution, as it will inform further research on its topic.

Many thanks for your assessment and for appreciating the value of the results. We are grateful for your constructive comments and the time you have spent reviewing this manuscript.

However, several points in the manuscript require further clarification.

Major comments:

* 1. As the Authors admit, the results of the paper are numeric rather than analytic, and hence they do not reveal if multiple boundaries give the same or just very similar reward. Nevertheless more numeric investigation of this question would be useful. For example in Figure 5a it is unclear if multiple sets of parameters exist OR there is just more sensitivity to theta than to alpha. To shed more light on this question, it would be good to plot R for $\alpha = \{-20, -10, 0, 10, 20\}$ for optimal thetas for a given alpha from Figure 5a with 100 times more iterations than in Figure 5a (which would be feasible as the plot would just include 5 points) and display also error bars showing standard error.

This is a very good point and also relates to concretely demonstrating some of the points raised by reviewer 1.

We have included a new figure (figure 6: cross sections through the reward landscapes) that does exactly this, and a substantial amount of new text in the section 'A degenerate set of decision boundaries yield close-to-optimal expected reward' covering the implications of the results in this figure.

* 2. Figure 6 is an important figure in this paper, because it aims at explaining the reason for the existence of multiple equally good boundaries, but at the moment it is difficult to read. The caption says: "Examples of qualitatively different degenerate decision boundaries, overlaid on the mean decision time and mean error" – I do not see the mean error on this figure. Have the Authors forgotten to include the panels with error? If so, please add them, or otherwise clarify how to read out the error from this figure. Also the colour scale in Figure 6 is difficult to read. In the text the Authors made an interesting observation: "decision times vary even along the flat boundaries (figure 6, yellow lines)", but this is very difficult to see. Can the figure be modified to make this surprising effect easier to see?

This figure has now become figure 7 in the revised version.

There was a mistake in the caption in referring to the decision error, which we have corrected – we originally had the errors shown, but on balance felt it confused the message that was more simply represented from the decision times.

Several improvements have been made to this figure that should hopefully make it a lot more legible now.

* 3. This manuscript sheds light on questions which have been puzzling me related to existing tests for decision making between multiple options. It has been already shown by Dragalia et al. (1999 IEEE Transactions on Information Theory) that there exist two tests MSPRTa and MSPRTb which both are asymptotically optimal in a limit of error going to 0. Since these tests are well known and often studied, it would be helpful to analyse them in the framework of this paper. I have been always wondering: Which one of them is really better for larger error rates and why do they become equivalent in the limit. I feel that the methodology developed in the manuscript can help answer these questions. First, it would be interesting to explicitly compare the shape of the boundaries of the two test and visualize them on the “triangle plots” (like Figure 6). MSPRTa corresponds to “flat boundaries”, and it would be interesting to see the shape for MSPRTb. Second, it would be nice to compare in simulations the rewards for the two tests. Third, does the asymptotic optimality relate to the fact that in the limit of error $\rightarrow 0$ the bounds become a point in the “triangle plots” for both tests, and therefore the tests become equivalent?

In answer to your question, we would say that McMillen and Holmes (2005) have compared the two MSPRT tests empirically and found the mean reaction times from both tests were the same at all error rates they examined (i.e. the tests perform equivalently)

In our case, for comparing the boundary shape, the relevant decision variables for the two tests do not occupy the same space, which would therefore make generating a figure for a boundary comparison on the same plot difficult. Our paper is also quite full, and this point is tangential to our main points, so we would prefer to treat this question as out of the scope of our study.

We do agree with your insight on the shape of the two decision planes resulting in same asymptotic optimality result is probably correct, but as we say would see this as the basis for a future study.

Minor comments:

* 1. Figure 4: It is not clear if all parameters have been optimized, or just theta. If the latter, please give the values of other parameters.

Yes, all parameters were optimized, which has been emphasised in a new caption to Fig 4: ”SAT curves for 3-choices optimized over all parameters.”

The parameters are not shown on this Figure 4 (where they are implicit) but on Figure 5 instead, which does show the alpha parameter as well as theta.

* 2. In Section “Implicit dynamics of optimal decision boundaries support both static and collapsing thresholds” the Authors wrote: “the boundary beliefs on choice selection, as generated by Monte-Carlo decision trajectories, are transformed to a log-likelihood representation (equation (2))”. It is not clear to me why this transformation is performed – could you please justify? (Equation 2 was introduced in the context of 2 alternatives, while this section is about 3 alternatives, so it is not clear how it is relevant).

We agree this was not clear. We have justified this transformation for visualization purposes: ‘The boundary beliefs are sorted by decision time, averaging over boundary values with identical decision times. We thus find time-dependent decision boundaries applied to evidence (figure 8), which for display purposes we represent using the log odds (equation 2).’

* 3. In Discussion the Authors say: “The implication is that the urgency signals reported in neural recordings of decision making tasks, such as those recorded in area LIP [7, 8], could in fact be the hallmark of a more complex decision rule, and that the signal as a distinct phenomenon may be misleading as to the true mechanism.” I feel this statement is too strong for 2 reasons: First, the urgency signals have also been observed for 2 alternatives (see Figure 4a in Churchland et al. 2006 Nature Neuroscience), while the theory presented by the Authors does not predict any varying boundaries for two options. Second, the experiments investigating urgency signals (e.g. by Churchland et al. 2006 Nature Neuroscience) involved trials with mixed difficulty, while the manuscript considers constant difficulty. As Authors write later, it has been demonstrated by Drugowitsch et al (2012 J Neurosci) that when evidence varies in difficulty, as in these experimental studies, then the urgency signals may indeed improve reward rate. Due to the differences in the assumptions of the manuscript and the experimental studies investigating urgency signals the statement cited above needs to be tempered.

Agreed. We have amended the sentence:

“The implication is that the urgency signals reported in neural recordings of decision-making tasks with more than two alternatives, such as those recorded in area LIP [7, 8], could in part be the result of a more complex decision rule.”

* 4. In section “Predictions”, the Authors say that their model predicts that initial activity should depend on the number of alternative choices. I feel it needs to be clarified that this prediction is not unique to the model described in the manuscript, but applies to a more general class of models in which the activities of integrators are normalized to represent the probabilities (e.g. Ditterich 2010 Frontiers Neurosci.; Bogacz & Larsen 2011 Neural Computation. The Authors may find interesting a comparison of this prediction with experimental data in a book chapter by Bogacz (2009, Optimal decision-making theories. In Handbook of reward and decision making, pp. 373-397. Academic Press).

This is good point. We have amended the last sentence the first paragraph in Predictions section:

“As such, we offer a variation of these predictions and so a means of distinguishing empirically between the nDRM and the class of models, including the one presented here, in which the activities of integrators are normalized to represent the probabilities (e.g. [44, 45]).”

* 5. On page 21: “For n choices, the decision space has C^n_3 faces” – I do not understand what it means, and had problems with following the text below. Please clarify.

We agree, and have revised this paragraph to hopefully make it clearer:

“We have parameterized in terms of 3D subspaces representing the outer extent of the decision space (figure 1) given by $P_l(t) + P_m(t) + P_n(t) = 1$, where $l \neq m \neq n$. So, the number of outer 3D subspaces within the decision space is the number of unique unordered combinations of three choices, which is denoted C_n^3 for n choices (one for $n=3$, three for $n=4$).”

We have also included specific examples for $n=3$ and $n=4$:

“Each outer subspace has a three-element belief vector $P(t)_\phi$ with elements $p(t)_i$, and the set of all such 3D subspace vectors is ΣC^3 : for $n=3$ this is set of one vector, so that $\Sigma C^3 = P(t)\{0,1,2\} = \{(p(t)_0, p(t)_1, p(t)_2)\}$; for $n=4$, this is a set of three vectors: $\Sigma C^3 = \{P(t)\{0,1,2\}, P(t)\{0,1,3\}, P(t)\{0,2,3\}, P(t)\{1,2,3\} = \{(p(t)_0, p(t)_1, p(t)_2), (p(t)_0, p(t)_1, p(t)_3), (p(t)_0, p(t)_2, p(t)_3), (p(t)_1, p(t)_2, p(t)_3)\}$ ”

And some additional text to aid with intuition:

“For example, for $n=4$, if one alternative has a zero probability the decision space and thresholds collapse to the $n=3$ case (the planes shown in figure 1), constraining how the decision boundaries intersect with the outer faces. Secondly, it introduces the simplest form of nonlinearity, as the product of all elements $P(t)_\phi$ parameterizes curved boundaries.”

“Summing these components over all subspaces generalises (14) to n choices, for which it is normalized by the number of 3D subspaces using the multiplier $1/C_n^3$.”

REVIEWER COMMENTS

Reviewer #1 (Remarks to the Author):

I appreciate that the authors revised the manuscript to relativize their findings in the context of past literature. I also appreciate them now making explicit that the degeneracy that they explore is not strictly mathematical, but rather encompasses practically "good enough" strategies. Unfortunately, the manuscript still contains numerous imprecisions - too many to identify them all individually. I give a few examples below, but encourage the authors to scrutinize all the manuscript's statements with respect to precision and clarity.

p2: "Collapsing boundaries are known to be optimal for value-based binary-choice tasks of mixed difficulty [24, 16]" - [24] relates to perceptual decision-making, not value-based decision making.

p2: "Almost all of the above-mentioned models (except [11]) assume independent, constant-valued decision boundaries for each choice accumulator. [...] accumulators, by which evidence accumulation and decision criteria are independent." - still unclear / imprecise. Commonly, models either perform independent evidence accumulation or have independent boundaries, but not both. Updating the posterior by Bayes' rule (as in the MSPRT), requires normalization across posteriors, making the accumulators depend on each other. LCA also has interacting accumulators. Only simple race models have both independent accumulators and boundaries, but those are severely limited, and don't match the data well. [11] considers both cases: independent accumulators and dependent boundaries (the non-network model), and interacting accumulators and independent boundaries (the network model).

p3: "The integration-to-threshold model samples evidence from the 'true' hypothesis until a decision boundary is reached." - usually, integration-to-threshold models rely on scalar evidence that is bounded by a scalar boundary. Once the evidence becomes vector-valued, the above statement becomes unclear. What is the boundary on in such circumstances? Please clarify.

p6: "Our interest is whether there are other complex boundary shapes that: [...] ii) can perform comparably to the curved optimal boundaries found in [11]." - I didn't find an exploration of ii) in the manuscript. There is no reason to assume that the parametrically curved boundaries used in the manuscript match those used in [11].

Fig 4: how do all the SATs compare to a flat boundary? That comparison seems to be missing throughout the whole manuscript, except for a brief mention in the context of Fig. 5 for $\alpha = 0$.

Fig 4: How do you plot the dashed line? Is it a piecewise linear curve that connects some averages? If yes, averages of what? If not, how is this curve generated? Same question applies to Fig 5, d-f.

p8: "However, the small differences between the SAT curves indicate that a wide range of boundary characteristics are still capable of generating near optimal decisions." - These small differences are hard/impossible to see from the figures themselves. Could you quantify them, or plot the data differently to make them more visible?

Fig 6: in my opinion, the text describing the results shown in Fig 6 still doesn't elaborate enough on the stochasticity resulting from Monte Carlo evaluation of the average reward, and how it relates to choosing what you consider close-to-optimal reward rates. Specifically, a larger number of MC simulations would reduce this stochasticity, and with it (potentially) the range of bound parameters that you might consider close-to-optimal. Furthermore, it might impact the qualitative comparison shown in Fig. 6. Thus, the results need to be relativized with respect to the chosen number of MC simulations. This also applies to the red lines shown in Fig. 6, whose distance would decrease with the number of MC simulations, which would quantitatively and qualitatively impact the drawn conclusions.

Fig. 6, red lines: in addition to the number of MC simulations, the red lines depend on the range of explored alphas. Would the same qualitative results hold if this range increases? Either way, it needs to be made clear that the statements regarding the red lines are relative to the chosen range of alphas.

p10: "This overlap decreases as W/c increases but does not diminish to zero, which would have indicated a single maximum, and instead supports a broad set of optimal parameters for all values of W/c " - doesn't your definition of optimality yield a broad set of optimal parameters, irrespective of if the "overlap" goes to zero? The above statement seems to not discriminate between your defined threshold of what you consider close-to-optimal, and stochasticity due to a limited number of MC simulations, but nonetheless appears to assign a large range of optimal parameters to the limited number of MC simulations only.

p11: "For every cost ratio W/c , the entire range of alpha and beta is represented in the optimal set" - you can only make this statement about the explored range of alpha and beta, not all possible alpha's and beta's. Maybe change to "the explored range of alpha and beta"?

p12: "whereas static thresholds should be used on single free response trials without deadlines [18]" - not sufficiently precise. Static thresholds are also adequate in repeated free-response trials of known difficulty. They are not adequate for single free response trials of mixed, a priori unknown, difficulty.

p13: "as a special case where evidence is independent over choices" - I am not sure I (or other readers) understand what this "evidence is independent over choices" means. Please elaborate.

Fig 8: additional info about the parameter regimes that yield the qualitatively different curves would be helpful. In particular, in which case do you find constant time-accuracy curves (i.e., last row)?

p15: "Our first main result [...]" - as you acknowledged in the revisions, this has been established previously, and so isn't a new result. Leftover from previous version?

p16: "For this reason, comparing the performance to single-valued flat boundaries for the nDRM is not comparable to the flat boundaries examined here." - This is not the only reason. Mixed difficulty, which demands collapsing boundaries, is another one.

p17: "A similar approximation and neural implementation would be possible with the non-linear boundaries presented in this paper." - I am not convinced that this is indeed the case. In particular, how would a combined convex/concave boundary (e.g., power, oscil in your manuscript) be implemented like this?

p17: "However, non-linear boundaries do not apply to 2AFC tasks" - please again clarify that here you are again referring to non-linearity in (accumulation) space, not in time.

p18: "However, our predictions differ due to the degeneracy of the decision boundaries. We predict that this manifold may vary from subject-to-subject or trial-to-trial while maintaining near optimal performance." - To me, it is unclear where this comes from. The accumulation manifold is relevant before hitting the boundary, so why and how does the boundary shape impact the nature of the neural manifold on which evidence accumulation occurs? Please clarify/elaborate.

Minor:

p6: "curve(Θ , α), power(Θ , α , β), and oscil(Θ , α , β)" - you are using lowercase theta and uppercase Θ throughout the manuscript to refer to the same parameter. Please stick to either lowercase or uppercase.

p10: "II) The red dashed lines comparing the overlap [...]" - this doesn't seem to be a complete sentence.

p12: "decision boundaries would shrink and the range of SATs narrow" - sentence fragment

p13: "(equation (2))." - second closing bracket missing.

Reviewer #2 (Remarks to the Author):

The Authors sufficiently modified the manuscript according to my comments, and I feel this paper would be a nice addition to the literature.

After re-reading the manuscript I only have one small comment.

I am confused by the relationship in Figure 4 between W/c and the optimal speed accuracy tradeoff. According to Eq 1, W is the weight of error, so I would expect that high W should result in high accuracy. By contrast high W/c ratio (yellow points) correspond in the figure to low accuracy. Could you please clarify why this is the case? Or is it a typo in a label?

Response for manuscript ‘Degenerate Boundaries for Multiple-Alternative Decisions’ Submitted to Nature Communications (NCOMMS-20-24534A)

Please find attached our revised manuscript and point-by-point response below.

In addition, we have worked very carefully through the text to correct further imprecisions and improve the writing quality. These do not affect the scientific content of the work, but significantly raise the quality of the paper.

To help track the changes across versions, two copies of the revised manuscript have been included: (1) highlighting the changes detailed in the point-by-point response; and (2) using a track-changes LaTeX package that shows all differences from the previous version. (Please note that the package does not fully track changes in equations and figure legends, just that a change has been made.)

Reviewer #1

I appreciate that the authors revised the manuscript to relativize their findings in the context of past literature. I also appreciate them now making explicit that the degeneracy that they explore is not strictly mathematical, but rather encompasses practically "good enough" strategies. Unfortunately, the manuscript still contains numerous imprecisions - too many to identify them all individually. I give a few examples below, but encourage the authors to scrutinize all the manuscript's statements with respect to precision and clarity.

You have made a very clear case that we have been imprecise in some details in the paper. Apologies for the extra work this has created for you in reviewing; we are grateful for your efforts to raise the quality of our paper. All authors have tried our best to carefully go through the paper to correct the points you raise and any additional imprecisions we can find. Our responses are given below and the text is highlighted in the paper.

In addition, the most experienced (last) author has thoroughly worked over the paper to improve the writing quality. These changes have not been highlighted as they are mostly minor changes of phrasing, but this has also raised the overall quality (e.g., the mathematical methods had many minor mistakes in notation, as can be seen by comparing versions.)

p2: "Collapsing boundaries are known to be optimal for value-based binary-choice tasks of mixed difficulty [24, 16]" - [24] relates to perceptual decision-making, not value-based decision making.

Corrected:

“Collapsing boundaries are known to be optimal for value-based binary-choice tasks [17] and perceptual tasks of mixed difficulty [18], as well as tasks where decisions are subject to a time deadline [19, 20].”

(Please note that citation numbers have changed because of the edits – e.g. now in order)

p2: "Almost all of the above-mentioned models (except [11]) assume independent, constant-valued decision boundaries for each choice accumulator. [...] accumulators, by which evidence accumulation and decision criteria are independent." - still unclear / imprecise. Commonly, models either perform independent evidence accumulation or have independent boundaries, but not both. Updating the posterior by Bayes' rule (as in the MSPRT), requires

normalization across posteriors, making the accumulators depend on each other. LCA also has interacting accumulators. Only simple race models have both independent accumulators and boundaries, but those are severely limited, and don't match the data well. [11] considers both cases: independent accumulators and dependent boundaries (the non-network model), and interacting accumulators and independent boundaries (the network model).

Corrected:

“Almost all of the above-mentioned models of decision making (except [6]) assume independent, constant-valued decision boundaries for each choice accumulator. For those models, the decision boundaries cannot depend on time or the state of other accumulators, so existing models with inter-acting accumulators, such as the MSPRT and leaky competing accumulator (LCA) models, limit that interaction to normalization [12] or lateral inhibition across accumulators [21], respectively. However, a core aspect of multiple-choice decision dynamics is that the interaction of accumulators or equivalent decision boundaries may have a non-trivial dependence on the belief over all choices, which can then act as belief-dependent gains on the choice evidence or as nonlinear decision boundaries. In particular, optimal boundaries for general multi-alternative decisions have been shown to be both time-dependent and nonlinear [6].”

p3: "The integration-to-threshold model samples evidence from the ‘true’ hypothesis until a decision boundary is reached." - usually, integration-to-threshold models rely on scalar evidence that is bounded by a scalar boundary. Once the evidence becomes vector-valued, the above statement becomes unclear. What is the boundary on in such circumstances? Please clarify.

Corrected:

“Usually, integration-to-threshold models rely on scalar evidence with a scalar decision boundary. In our case, the evidence will be a vector with boundaries that are hyper-surfaces in a vector space, which is detailed in the next section.”

p6: "Our interest is whether there are other complex boundary shapes that: [...] ii) can perform comparably to the curved optimal boundaries found in [11]." - I didn't find an exploration of ii) in the manuscript. There is no reason to assume that the parametrically curved boundaries used in the manuscript match those used in [11].

Corrected:

“Here we ask whether there are other complex boundary shapes that improve performance over the flat boundary case, and whether the greater freedom to choose nonlinear boundaries has other consequences for the decision making.”

Fig 4: how do all the SATs compare to a flat boundary? That comparison seems to be missing throughout the whole manuscript, except for a brief mention in the context of Fig. 5 for $\alpha = 0$.

Clarified (p9):

p8: “Flat boundaries ($\alpha = 0$) are contained within these parameterizations, and so the SAT curves for all three boundary functions closely resemble the flat-boundary case.”

Fig 5 caption: “Note that the $\alpha = 0$ (cyan) sections are flat decision boundaries.”

p12: "The flat-boundary case ($\alpha=0$) is optimal for each of the degenerate sets, with the optimal θ then a single value that lies within the broadened range when α is non-zero."

Fig 4: How do you plot the dashed line? Is it a piecewise linear curve that connects some averages? If yes, averages of what? If not, how is this curve generated? Same question applies to Fig 5, d-f.

Clarified:

Fig 4 caption: "The average SAT curve (dashed line) is a piecewise-linear curve from the mean decision time and error over all three parameterizations for each c/W value (see methods)."

Fig 5 caption: "with the black dashed line showing the mean SAT curve from figure 4."

p8: "However, the small differences between the SAT curves indicate that a wide range of boundary characteristics are still capable of generating near optimal decisions." - These small differences are hard/impossible to see from the figures themselves. Could you quantify them, or plot the data differently to make them more visible?

This is not what we meant to imply; this text has been rewritten:

"One difference between the three cases is their relative spreads: the curve(θ,α) parameterization yields the tightest SAT curve, followed by the $\text{oscil}(\theta,\alpha,\beta)$ SAT curve, and finally the $\text{power}(\theta,\alpha,\beta)$ SAT curve is the thickest. One might therefore consider that the curve parameterization qualifies as the 'best' SAT curve, which is consistent with previous work since it describes the shape of the optimal boundary found for the 3AFC case in [6, 12]. However, all parameterizations closely follow a single mean SAT curve (black dashed lines), so a wide range of boundary characteristics give near-optimal decisions. Moreover, each value of c/W has multiple points spread along the same curve, so the SAT can be satisfied by multiple optimised boundaries even within the same parameterization."

Fig 6: in my opinion, the text describing the results shown in Fig 6 still doesn't elaborate enough on the stochasticity resulting from Monte Carlo evaluation of the average reward, and how it relates to choosing what you consider close-to-optimal reward rates. Specifically, a larger number of MC simulations would reduce this stochasticity, and with it (potentially) the range of bound parameters that you might consider close-to-optimal. Furthermore, it might impact the qualitative comparison shown in Fig. 6. Thus, the results need to be relativized with respect to the chosen number of MC simulations. This also applies to the red lines shown in Fig. 6, whose distance would decrease with the number of MC simulations, which would quantitatively and qualitatively impact the drawn conclusions.

Fig. 6, red lines: in addition to the number of MC simulations, the red lines depend on the range of explored alphas. Would the same qualitative results hold if this range increases? Either way, it needs to be made clear that the statements regarding the red lines are relative to the chosen range of alphas.

p10: "This overlap decreases as W/c increases but does not diminish to zero, which would have indicated a single maximum, and instead supports a broad set of optimal parameters for all values of W/c " - doesn't your definition of optimality yield a broad set of optimal parameters, irrespective of if the "overlap" goes to zero? The above statement seems to not discriminate between your defined threshold of what you consider close-to-optimal, and stochasticity due to a limited number of MC simulations, but nonetheless appears to assign a large range of optimal parameters to the limited number of MC simulations only.

These points have been covered collectively in a new paragraph:

“These three observations all support the effective degeneracy of optimized decision boundaries within the parameterizations. Observation (I) shows that for small c/W , the underlying structure of the reward landscape appears degenerate with sections almost entirely overlapping (figure 6, top right). As c/W increases, a shallow structural maximum become apparent (figure 6, bottom right). Observation (II) shows significant overlap in the close-to-optimal region even across the apparent structural maximum (figure 6, bottom right). Observation (III) shows directly that there is an effective degeneracy. One could question whether different sections through the reward landscape would change these observations, as figure 6 depends on the range and discretization of α . Our range of α covers the entire range of boundary shapes shown in figure 2, including flat boundaries, and because of the gradual variation across sections we would not expect further structure from a finer discretization. We also expect that using more Monte Carlo samples for each cross section would not change the results, as the means and spreads shown in the sections in figure 6 appear to be good estimates of the distributions of average rewards (e.g. by their smooth variation with θ and unitary maxima).”

Please note that choosing more MC samples would not change the underlying distribution, only the accuracy of the estimated mean and spread (lines in figure 6). i.e. we would not expect the bounds to change much with more sampling, and likewise the red lines showing the overlap between spreads at the peaks.

p11: "For every cost ratio W/c , the entire range of alpha and beta is represented in the optimal set" - you can only make this statement about the explored range of alpha and beta, not all possible alpha's and beta's. Maybe change to "the explored range of alpha and beta"?

Corrected:

“For every cost ratio c/W , the entire explored range of α is represented in the optimal set ”

p12: "whereas static thresholds should be used on single free response trials without deadlines [18]" - not sufficiently precise. Static thresholds are also adequate in repeated free-response trials of known difficulty. They are not adequate for single free response trials of mixed, a priori unknown, difficulty.

Corrected:

“whereas static thresholds are appropriate for single free-response trials without deadlines and repeated free-response trials of known difficulty; however, static thresholds are not adequate for single free-response trials of mixed, a priori unknown difficulty [24].”

p13: "as a special case where evidence is independent over choices" - I am not sure I (or other readers) understand what this "evidence is independent over choices" means. Please elaborate.

Clarified:

“Flat decision boundaries can therefore be interpreted as a special case where the boundary on the belief in one choice to cross its threshold is independent of the beliefs of the other choices; however, even then, the mean decision time and mean error have a spatial structure along these boundaries (figure 7).”

Fig 8: additional info about the parameter regimes that yield the qualitatively different curves would be helpful. In particular, in which case do you find constant time-accuracy curves (i.e., last row)?

“ and static thresholds with flat boundaries (e.g. figure 7a, yellow line).”

p15: "Our first main result [...]" - as you acknowledged in the revisions, this has been established previously, and so isn't a new result. Leftover from previous version?

Corrected:

“Firstly, evidence accumulation takes ...”

p16: "For this reason, comparing the performance to single-valued flat boundaries for the nDRM is not comparable to the flat boundaries examined here." - This is not the only reason. Mixed difficulty, which demands collapsing boundaries, is another one.

Clarified:

“For this reason, and because [6] uses mixed difficulties which demands collapsing boundaries, comparing the performance to flat boundaries for the nDRM is not comparable to the flat boundaries examined here.”

p17: "A similar approximation and neural implementation would be possible with the non-linear boundaries presented in this paper." - I am not convinced that this is indeed the case. In particular, how would a combined convex/concave boundary (e.g., power, oscil in your manuscript) be implemented like this?

Clarified:

“An interesting question is whether a similar approximation and neural implementation can be implemented with the nonlinear boundaries presented here.”

“However, it is unclear whether a recurrent network (as in [6]) exists to transform the decision variables for the more complex power and oscil boundaries. If it is possible to represent the nonlinearities within a larger recurrent network, one would obtain a local, single-valued decision rule applied independently to each accumulator.”

p17: "However, non-linear boundaries do not apply to 2AFC tasks" - please again clarify that here you are again referring to non-linearity in (accumulation) space, not in time.

Clarified:

“However, boundaries that are nonlinear in the evidence (but linear in time) do not apply to 2AFC tasks.”

p18: "However, our predictions differ due to the degeneracy of the decision boundaries. We predict that this manifold may vary from subject-to-subject or trial-to-trial while maintaining near optimal performance." - To me, it is unclear where this comes from. The accumulation manifold is relevant before hitting the boundary, so why and how does the boundary shape impact the nature of the neural manifold on which evidence accumulation occurs? Please clarify/elaborate.

Clarified:

“However, there may be further subtlety in our case due to the effective degeneracy of the decision boundaries. The section ‘Implicit dynamics of optimal decision boundaries support both static and collapsing thresholds’ showed that the decision variable can be transformed into a form that gives apparent temporal structure in the threshold (figure 8), dependent on the particular nonlinear boundary within the degenerate set. Our expectation is that this subtlety in the threshold dynamics may also manifest in that the accumulation manifold may appear to vary from subject-to-subject or trial-to-trial while maintaining near optimal performance (given that any point in the accumulation manifold could also be on some decision boundary).”

Minor:

p6: "curve(Theta, alpha), power(Theta, alpha, beta), and oscil(Theta, alpha, beta)" - you are using lowercase theta and uppercase Theta throughout the manuscript to refer to the same parameter. Please stick to either lowercase or uppercase.

Corrected. Now all lower case (apart from Eq 2 where a distinction is necessary).

p10: "II) The red dashed lines comparing the overlap [...]" - this doesn't seem to be a complete sentence.

Rewritten:

“The spread of the average rewards at the peak of each section overlaps (red dashed lines), decreasing as c/W increases but not diminishing to zero.”

p12: "decision boundaries would shrink and the range of SATs narrow" - sentence fragment

Corrected.

p13: "(equation (2)." - second closing bracket missing.

Corrected

Reviewer #2

The Authors sufficiently modified the manuscript according to my comments, and I feel this paper would be a nice addition to the literature.

After re-reading the manuscript I only have one small comment:

I am confused by the relationship in Figure 4 between W/c and the optimal speed accuracy tradeoff. According to Eq 1, W is the weight of error, so I would expect that high W should result in high accuracy. By contrast high W/c ratio (yellow points) correspond in the figure to low accuracy. Could you please clarify why this is the case? Or is it a typo in a label?

Thank you, you found a mistake in our use of W/c throughout the paper. We had the two quantities flipped in the text compared to the figures, and have now corrected it to refer to c/W instead.

REVIEWER COMMENTS

Reviewer #1 (Remarks to the Author):

I appreciate the author's extended revisions of the manuscript. While they have improved the manuscript, I still have a few points that I am unclear about:

(page numbers are for the version that tracks additions and deletions)

p2: "For those models, the decision boundaries cannot depend on time or the state of other accumulator, so existing models with interacting accumulators, [...], limit that interaction to normalization [...]" - How does the second part (after 'so ...') follow from the first? I don't see how the shape of the decision boundaries constraints how accumulators can interact with each other before reaching these boundaries.

p10: "For nAFCs with $n > 2$, these parameters [...] set the boundary function in Eq. (3), which in effect would over-specify the SAT curve if it remains a curve of zero thickness" - What do you mean with "over-specify the SAT curve"? Why would the SAT curve be "over-specified"?

p14: paragraph starting with "How can a small range of reward values produce a range of speed-accuracy tradeoffs?": I am not sure I understand the argument here. For each set of boundary parameters, there exists, by definition, one $E[T]$ and one $E[e]$ (irrespective of the complexity of the boundary). The range of decision times and errors in Fig. 5d-f appear to arise from varying the parameters (i.e., choosing them from the set of parameters that yield near-optimal performance). You instead seem to claim in this paragraph that the different $E[T]$ and $E[e]$ arise from a different effect. At least, that's what your statement "It appears that the reward structure (Eq. 4) is under-determined for nAFCs with $n > 2$ " suggests. You could equally find a range of $E[T]$ and $E[e]$ for 2AFCs if you choose all the boundary heights that yield near-optimal (rather than strictly optimal) performance, which seems to contradict your statement.

p20: "They find that while both nonlinearity and temporal dynamics improve performance, nonlinearity has a greater effect" - I assume that the authors refer to the "constraint alone" condition Fig. 3 of [18], which the authors call "nonlinearity". To my understanding, this constraint actually performs the probability normalization, and so doesn't only induce the boundary's curviness. As far as I can tell (and as the authors point out later in the manuscript) [18] doesn't consider the effects of probability normalization and curved boundaries in isolation. Thus, from [18] we can't tell how much of an effect the curved boundaries had on the final performance.

Response for manuscript ‘Degenerate Boundaries for Multiple-Alternative Decisions’ Submitted to Nature Communications (NCOMMS-20-24534B-Z)

Please find attached our revised manuscript (with highlighted changes) and point-by-point response below.

Once again, we would like to express our gratitude for the careful and expert reviewing.

Reviewer #1:

I appreciate the author's extended revisions of the manuscript. While they have improved the manuscript, I still have a few points that I am unclear about:

(page numbers are for the version that tracks additions and deletions)

p2: "For those models, the decision boundaries cannot depend on time or the state of other accumulator, so existing models with interacting accumulators, [...], limit that interaction to normalization [...]" - How does the second part (after 'so ...') follow from the first? I don't see how the shape of the decision boundaries constraints how accumulators can interact with each other before reaching these boundaries.

This was a typo: it should have been ‘and` rather than ‘so’ – now corrected.

p10: "For nAFCs with $n > 2$, these parameters [...] set the boundary function in Eq. (3), which in effect would over-specify the SAT curve if it remains a curve of zero thickness" - What do you mean with "over-specify the SAT curve"? Why would the SAT curve be "over-specified"?

Apologies, this was not clear – we have changed this to have a more precise phrasing:

“For nAFCs with $n > 2$, these are complex boundary functions (equation (3)) with sets of parameters. If the SAT curve is truly a curve, rather than a region, then multiple parameter combinations would give the same speed-accuracy tradeoff, since a curve requires just one implicit parameter.”

p14: paragraph starting with "How can a small range of reward values produce a range of speed-accuracy tradeoffs?": I am not sure I understand the argument here. For each set of boundary parameters, there exists, by definition, one $E[T]$ and one $E[e]$ (irrespective of the complexity of the boundary). The range of decision times and errors in Fig. 5d-f appear to arise from varying the parameters (i.e., choosing them from the set of parameters that yield near-optimal performance). You instead seem to claim in this paragraph that the different $E[T]$ and $E[e]$ arise from a different effect. At least, that's what your statement "It appears that the reward structure (Eq. 4) is under-determined for nAFCs with $n > 2$ " suggests. You could equally find a range of $E[T]$ and $E[e]$ for 2AFCs if you choose all the boundary heights that yield near-optimal (rather than strictly optimal) performance, which seems to contradict your statement.

We have rephrased this paragraph to not sound like we are contradicting ourselves, but rather to provide more explanation:

p11 (formerly p14): “How can a small range of reward values produce a broad range of speed-accuracy tradeoffs? The breadth of speed-accuracy tradeoff values produced by complex decision boundaries is explained by a range of near-optimal threshold parameter values. Then, given cost values W and c , the rewards

$$E(r_{\max}) = -W E(e) - c E[T], \quad e = \{0, 1\}, \quad (\text{correct/incorrect decision}) \quad (4)$$

have a freedom for each $E(r_{\max})$ in trading off the expected error $E(e)$ and expected decision time $E[T]$. Thus, the same expected reward may be attained by boundaries with different combinations of $E(e)$ and $E[T]$. Hence, the set of close-to-optimal decision boundaries yield the range of $(E(e), E[T])$ solutions shown in figures 5d-f.”

p20: "They find that while both nonlinearity and temporal dynamics improve performance, nonlinearity has a greater effect" - I assume that the authors refer to the "constraint alone" condition Fig. 3 of [18], which the authors call "nonlinearity". To my understanding, this constraint actually performs the probability normalization, and so doesn't only induce the boundary's curviness. As far as I can tell (and as the authors point out later in the manuscript) [18] doesn't consider the effects of probability normalization and curved boundaries in isolation. Thus, from [18] we can't tell how much of an effect the curved boundaries had on the final performance.

We don't think this sentence is needed to make our point, and so have deleted it to not mislead about ref. [7]. We considered including more detail but felt it complicated matters.

(Please note, it looks like you meant to reference [7] not [18].)